# Label-free chemical imaging of cytochrome P450 activity by Raman microscopy

Menglu Li [1,2,6], Yasunori Nawa[1,2,6], Seiichi Ishida[1,3,4], Yasunari Kanda [1,4], Satoshi Fujita [1,2✉] & Katsumasa Fujita [1,2,5✉]

Although investigating drug modulation of cytochrome P450 (CYP) activity under physiological conditions is crucial in drug development to avoid severe adverse drug reactions, the current evaluation approaches that rely on the destructive and end-point analysis can be misleading due to invasive treatments and cellular heterogeneity. Here, we propose a non-destructive and high-content method for visualizing and quantifying intracellular CYP activity under drug administration by Raman microscopy. The redox-state and spin-state sensitive Raman measurement indicated that the induced CYPs in living hepatocytes were in oxidized and low-spin state, which is related to monooxygenase function of CYP. Moreover, glycogen depletion associated with CYP induction was simultaneously observed, indicating a relevant effect on glucose metabolism. By deciphering the overall changes in the biochemical fingerprints of hepatocytes, Raman microscopy offers a non-destructive and quantitative chemical imaging method to evaluate CYP activity at the single-cell level with the potential to facilitate future drug development schemes.

[1] AIST-Osaka University Advanced Photonics and Biosensing Open Innovation Laboratory, National Institute of Advanced Industrial Science and Technology (AIST), 2-1 Yamadaoka, Suita, Osaka 565-0871, Japan. [2] Department of Applied Physics, Osaka University, 2-1 Yamadaoka, Suita, Osaka 565-0871, Japan. [3] Division of Applied Life Science, Graduate School of Engineering, Sojo University, 4-22-1, Ikeda, Nishi-ku, Kumamoto 860-0082, Japan. [4] Division of Pharmacology, National Institute of Health Sciences, Kawasaki, Kanagawa 210-9501, Japan. [5] Institute for Open and Transdisciplinary Research Initiatives, Osaka University, 2-1 Yamadaoka, Suita, Osaka 565-0871, Japan. [6] These authors contributed equally: Menglu Li, Yasunori Nawa. ✉email: s-fujita@aist.go.jp; fujita@ap.eng.osaka-u.ac.jp

Cytochrome P450s (CYPs), a superfamily of heme-containing enzymes expressed mainly in the liver, play a vital role in the oxidative biotransformation and elimination of drugs[1]. Among the 57 human CYPs, CYP3A4 is the most abundant type and is involved in the metabolism and clearance of almost half of administrated human drugs[2]. Modulation of CYP activity via induction or inhibition by drugs often leads to clinically significant drug–drug interactions (DDI) that can cause unanticipated adverse drug reactions (ADR) or therapeutic failures[3–5]. In addition, severe inhibition or induction of CYP has led to the withdrawal of drugs from the market causing time and financial losses[6]. To minimize DDI and reduce post-market withdrawal of drugs, investigation of the potent modulation of drugs on CYP activity is a requisite procedure before drug approval[7,8].

Current evaluation techniques of CYP activity rely on end-point analysis using invasive methods, such as mass spectrometry, fluorescence, and luminescence-based activity assays[9–11], which hinders comprehensive study at the cellular level. Although fluorescent and luminescent probes have been developed to image CYP activity in living cells[12–14], low biocompatibility and high cytotoxicity limit the application of most developing probes in living cells[10]. Moreover, the conventional methods evaluate the modulation of CYP activity by measuring the catalytic products[9–11]. Other factors have been demonstrated to impact the catalytic products, such as deficiency in the redox partners (NADPH-cytochrome P450 reductase or cytochrome $b_5$)[15,16] and inhibition of drug transporters[17]. Whether the impact is directly on CYP itself or other associating factors is difficult to clarify. Therefore, a method targeted on CYP enzyme independent from molecular probes will facilitate the assessment of CYP activity in the most natural conditions without concerns from associating factors.

In the present study, we reported Raman microscopy as a non-destructive method for detection and imaging of CYP activity in living hepatocytes without labeling. Raman microscopy has been used for label-free observation of biological molecules because it can directly detect molecular vibrations given by the intrinsic chemical structure of molecules[18]. The label-free format of Raman microscopy makes it a valuable tool for the evaluation and visualization of living biological samples under physiological conditions, which is particularly important in the field of biomedicine[19–21]. Live cell imaging by Raman microscopy has been achieved with a spatial resolution comparable to that of laser scanning confocal microscopy[22–24]. Also, Raman microscopy has been widely used in tissues in vivo or ex vivo for disease diagnosis[25–27]. The catalytically functional CYPs contain a heme prosthetic group, in which heme iron and porphyrin provide specific spectroscopic markers, allowing for the monitoring of chemical transformations and transitions by Raman spectroscopy[28–30]. The application of Raman microscopy to CYP studies allows visualization of CYP activity and the spatial distribution in individual cells without labeling. Differentiated HepaRG cells, a widely used hepatic cell line that expresses CYPs at levels comparable to primary hepatocytes[31,32], as well as human primary hepatocytes, were observed by Raman microscopy. Here, we showed that drug-induced modulation of hepatocyte-specific CYP activity and its associated effects on glucose metabolism can be assessed by Raman microscopy, providing an information-rich analysis method of drug effects without interfering with cell homeostasis.

## Results
**Visualization of CYP induction by Raman microscopy.** Rifampicin (RIF), a well-known CYP inducer, especially for CYP3A4, was applied to HepaRG cell culture to induce the expression of CYPs[33], and hyperspectral Raman images were captured. HepaRG

cells without RIF treatment were taken as controls. After comparing the Raman spectrum of the hepatocytes with RIF treatment (Fig. 1a), we found the most significant difference in the appearance of Raman bands at $1370\,\mathrm{cm}^{-1}$ and $1636\,\mathrm{cm}^{-1}$ (inserts, in Fig. 1a). From the previous resonance Raman studies of the CYP protein, substrate binds to CYPs at the ferric and low-spin (LS) resting state and triggers the CYP catalytic cycle. $1370\,\mathrm{cm}^{-1}$ and $1636\,\mathrm{cm}^{-1}$ were generally used as oxidation and LS state marker of CYP, respectively[28,30,34]. The appearance of $1370\,\mathrm{cm}^{-1}$ and $1636\,\mathrm{cm}^{-1}$ peaks after RIF treatment may suggest that the induced CYP is at an oxidized and LS state.

Raman imaging enables visualization of the spatial distribution of molecules at the target wavenumbers. The cytosolic distribution of $1370\,\mathrm{cm}^{-1}$ and $1636\,\mathrm{cm}^{-1}$ after CYP induction is shown in Fig. 1b, compared with peaks at $600\,\mathrm{cm}^{-1}$ (reduced heme $c$, mainly cytochrome $c$ or cyt $c$), $675\,\mathrm{cm}^{-1}$ (reduced heme $b$), and $1000\,\mathrm{cm}^{-1}$ (phenylalanine). A clear cytoplasm contrast appeared at $1370\,\mathrm{cm}^{-1}$ and $1636\,\mathrm{cm}^{-1}$ after RIF treatment. Since cyt $c$ localizes in the mitochondria, the increasing cytoplasm contrast after RIF treatment would not come from $c$-type heme but rather $b$-type heme.

In hepatocytes, CYP3A4 and cytochrome $b_5$ (cyt $b_5$) are the most abundant heme $b$ proteins associated with drug metabolism and electron transfer, respectively[35–37]. Therefore, we performed immunofluorescence staining for CYP3A4 and cyt $b_5$ after Raman observation. Figure 1c shows the immunofluorescence staining of CYP3A4 and cyt $b_5$ at the same position as the Raman observation. The fluorescent signals of CYP3A4 increased after RIF induction, while the signals of cyt $b_5$ remained constant, consistent with the western blotting results (Fig. 1d). After RIF induction, the Raman signal of $1370\,\mathrm{cm}^{-1}$ and $1636\,\mathrm{cm}^{-1}$ increased, similar to the changes observed in the fluorescent signal of CYP3A4 staining (Fig. 1c) and CYP3A4 activity assay (Fig. 1e). These results suggest that the increase of Raman shifts at $1370\,\mathrm{cm}^{-1}$ and $1636\,\mathrm{cm}^{-1}$ can be associated with the induction of CYP3A4 by RIF. Although RIF is a strong CYP3A4 inducer, other CYPs, like 2B6, were also elevated in HepaRG cells when treated with RIF (Supplementary Fig. 1). Since the heme $b$ core is the same in all CYP subtypes, other CYP types should also contribute to the arise of Raman peaks at $1370\,\mathrm{cm}^{-1}$ and $1636\,\mathrm{cm}^{-1}$. Therefore, we treated HepaRG cells with DMSO and omeprazole, to induce all CYPs and CYP1A2, respectively. The significant increase in $1370\,\mathrm{cm}^{-1}$ and $1636\,\mathrm{cm}^{-1}$ was revealed in DMSO-treated cells but not omeprazole (Supplementary Fig. 2). Since the same Raman shifts were detected in DMSO-treated cells, the Raman shifts at $1370\,\mathrm{cm}^{-1}$ and $1636\,\mathrm{cm}^{-1}$ can be assigned to several CYPs including CYP3A4. Although the treatment with OME elevated the expression of 1A2 for several folds, this amount is still undetectable by Raman microscopy possibly due to the low basal expression of CYP1A2 in HepaRG cells (Supplementary Fig. 2)[32,38].

Primary human hepatocytes (PHHs) are widely used in drug metabolism and screening study due to high basal expression of CYPs[39]. Therefore, we measured PHHs by Raman microscopy and compared with HepaRG cells. PHHs and HepaRG cells were observed right after cells have fully adhered to the collagen-coated surface. The Raman shifts at $1370\,\mathrm{cm}^{-1}$ and $1636\,\mathrm{cm}^{-1}$ were detected both in PHHs and HepaRG cells (Supplementary Fig. 3a). The time-course observation demonstrated the decreasing tendency of $1370\,\mathrm{cm}^{-1}$ and $1636\,\mathrm{cm}^{-1}$ after plating (Supplementary Fig. 3b), which was consistent with the immunostaining of CYP3A4 (Supplementary Fig. 3c). The expression of CYP in PHHs rapidly decreased once after plating, this known feature of primary hepatocytes was monitored by Raman microscopy[40]. Besides $1370\,\mathrm{cm}^{-1}$ and $1636\,\mathrm{cm}^{-1}$, we found dominant peaks at $1157\,\mathrm{cm}^{-1}$ and $1512\,\mathrm{cm}^{-1}$ (Supplementary Fig. 3a, indicated by asterisks), which can be assigned to

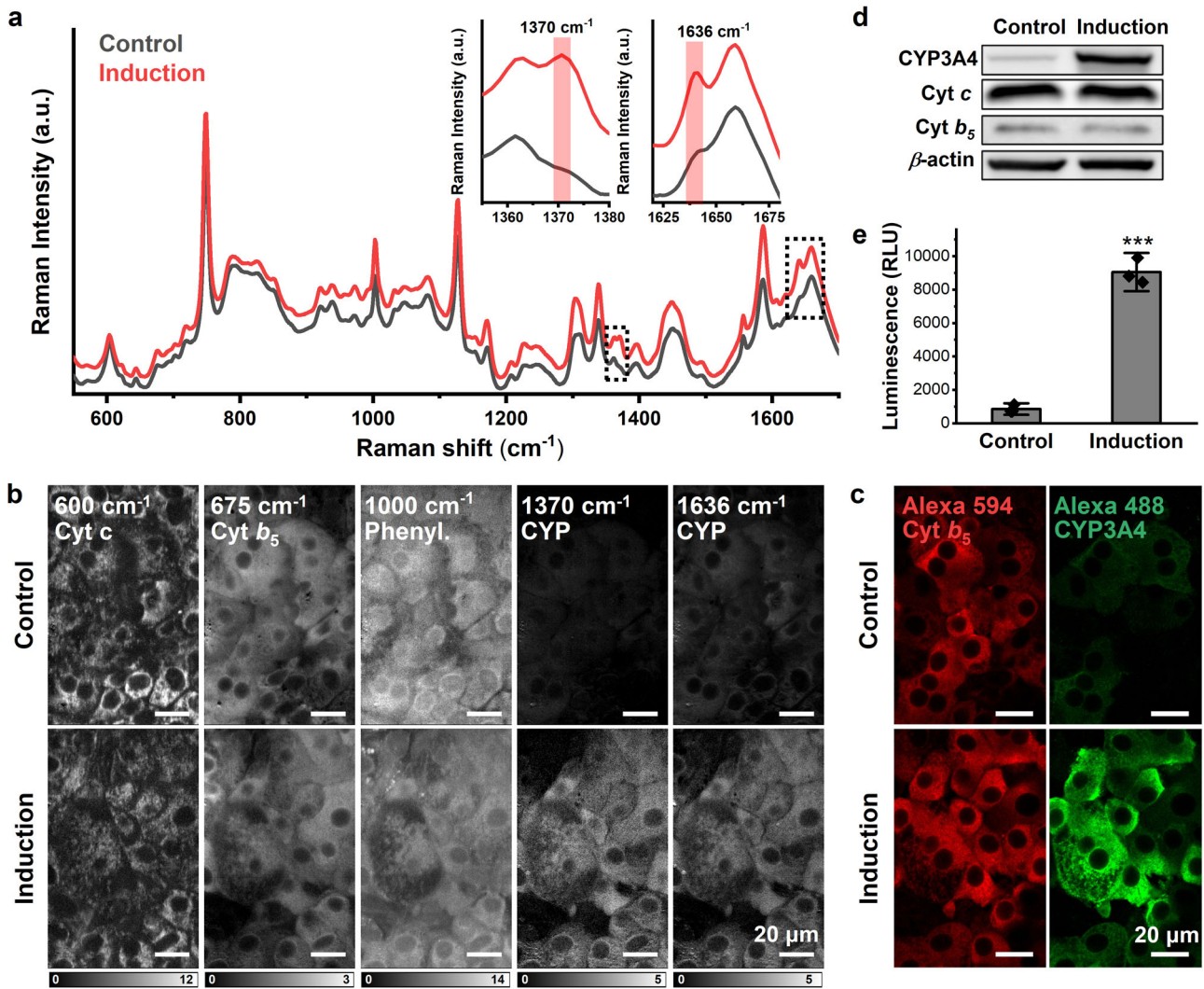

**Fig. 1 Measurement of CYP induction by Raman microscopy. a** Average Raman spectra of HepaRG cells treated with (red) or without (gray) rifampicin (RIF). The Raman shifts of 1370 cm$^{-1}$ and 1636 cm$^{-1}$ increased after treatment with RIF for 48 h; enlarged in the insets. **b** Reconstructed Raman images at 600 cm$^{-1}$, 675 cm$^{-1}$, 1000 cm$^{-1}$, 1370 cm$^{-1}$, and 1636 cm$^{-1}$, which can be assigned to cyt $c$, cyt $b_5$, phenylalanine, oxidized CYPs, and LS CYPs, respectively. The comparison of these cellular components demonstrated that 1370 cm$^{-1}$ and 1636 cm$^{-1}$ increased after CYP induction. Scale bars, 20 μm. **c** Immunofluorescence staining of cyt $b_5$ and CYP3A4 at the same positions as Raman measurement. Scale bars, 20 μm. **d** Western blots of CYP3A4, cyt $c$, cyt $b_5$, and β-action. **e** CYP3A4 activity assay indicates CYP induction after RIF treatment. Error bars indicate SD between triplicates. ***$P < 0.001$.

carotenoids stored in the liver[41]. Using Raman microscopy, we confirmed the same vibrational modes coming from CYPs in both PHHs and HepaRG cells. On the other hand, the difference between these cell lines was also clarified, demonstrating the potential of label-free Raman in identifying features of different cell lines based on their intrinsic cellular components.

Our results indicate that the oxidized and LS form of CYPs is directly related to the induction in living hepatocytes, which supports the conventional belief that the ferric and LS state is the starting point of the CYP catalytic cycle[42,43]. The redox state of CYP has been studied intensively in a reconstituted protein system but not in vitro cells by Raman spectroscopy. Johnston *et al.* investigated the redox state of human recombinant CYPs expressed in *E. coli* and living rat hepatocytes using Fe(II)-CO versus Fe(II) difference spectroscopy and reported that both ferrous and ferric CYPs existed in vitro[44]. However, the CO difference spectrum measurement remains a concern for interrupting the natural redox state of CYP[45]. Using Raman microscopy, we directly visualized cellular CYPs without external labels and demonstrated that ferric CYPs with LS were the major form after induction in living HepaRG cells.

**Non-destructive detection of hepatocyte-specific response to CYP induction.** Non-invasive characterization of different cell populations in an in vitro model could pave the way for the evaluation of cell type-specific and subsequent intercellular responses after drug administration. Differentiated HepaRG cells are known to present two morphologically and functionally distinct cell populations, including hepatocytes and biliary cells[46]. Hepatocytes form colonies and are surrounded by flat biliary cells. Raman spectra of hepatocyte colonies and biliary monolayer were collected in HepaRG cell without induction. Clear differences were found in the Raman spectra of the two cell populations (Fig. 2a). The Raman spectra of biliary cells were dominated by resonance peaks from reduced cyt $c$ (600 cm$^{-1}$, 640 cm$^{-1}$, 750 cm$^{-1}$, 1130 cm$^{-1}$, 1313 cm$^{-1}$, and 1585 cm$^{-1}$), protein (1000 cm$^{-1}$, phenylalanine), and lipids (1450 cm$^{-1}$, CH$_2$

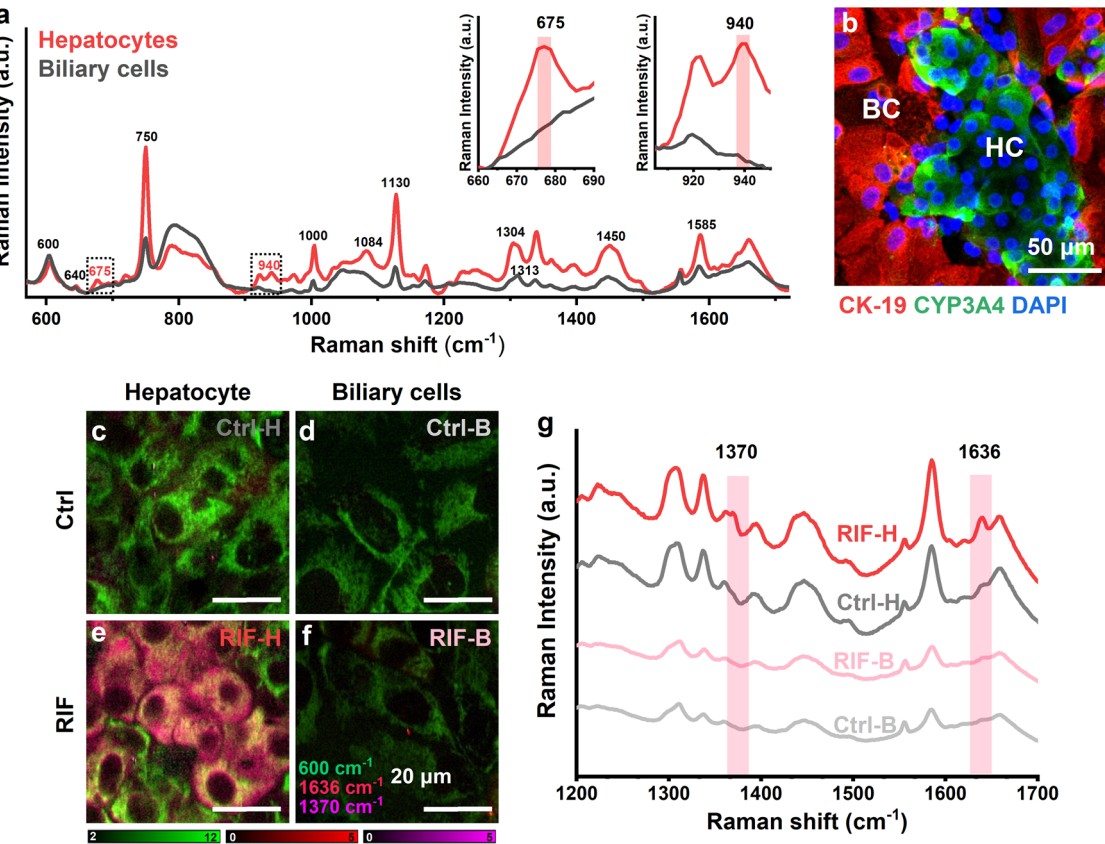

**Fig. 2 Hepatocyte-specific response to CYP induction was monitored by Raman microscopy. a** Raman spectra of hepatocytes (red) and biliary cells (gray) without CYP induction. **b** Immunofluorescence staining of CYP3A4 (green) and CK-19 (red) with nuclear counterstain by DAPI. Hepatocyte cells (HC) are CYP3A4 positive; while biliary cells (BC) are CK-19 positive. Scale bar, 50 μm. Reconstructed Raman images of control hepatocytes (**c** Ctrl-H), control biliary cells (**d** Ctrl-B), RIF-treated hepatocytes (**e** RIF-H), and RIF-treated biliary cells (**f** RIF-B) show the specific response of hepatocytes to RIF induction. Scale bars, 20 μm. **g** Average spectra of Ctrl-H (gray), Ctrl-B (light gray), RIF-H (red), and RIF-B (light red).

deformation) bands. In contrast, hepatocytes (without induction) preserved all the above-mentioned Raman shifts with higher intensities, indicating that these biomolecules are more abundant in hepatocytes. Furthermore, Raman shifts corresponding to reduced $b$-type cytochrome (675 cm$^{-1}$ and 1304 cm$^{-1}$) and glycogen (940 cm$^{-1}$ and 1084 cm$^{-1}$) appeared in hepatocytes. A comparison of representative 675 cm$^{-1}$ and 940 cm$^{-1}$ peaks is shown in the insets of Fig. 2a. The reconstructed Raman images comparing hepatocytes and biliary cells are shown in Supplementary Fig. 4a. The immunostaining images of cyt $b_5$ at the same position of Raman observation indicate that more cyt $b_5$ is expressed in hepatocytes than in biliary cells (Supplementary Fig. 4b).

After RIF treatment, expression of CYP3A4 was found in hepatocyte region but not in biliary cells by immunostaining of CYP3A4 (Fig. 2b, green). The staining of CK-19, a biliary marker, was performed together to identify biliary cells (Fig. 2b, red). The reconstructed images at 600 cm$^{-1}$, 1370 cm$^{-1}$, and 1636 cm$^{-1}$ are shown to reveal the localization of cyt $c$, oxidized CYPs, and LS CYPs, respectively (Fig. 2c–f). Hepatocytes showed a clear cytoplasm contrast of 1370 cm$^{-1}$ and 1636 cm$^{-1}$ after RIF induction (Fig. 2e), whereas only mitochondrial contrasts at 600 cm$^{-1}$ were observed in biliary cells (Fig. 2f), indicating that the occurrences of 1370 cm$^{-1}$ and 1636 cm$^{-1}$ were more specific to hepatocytes. This change in biochemical components was clearly observed in the comparison of Raman spectra (Fig. 2g). It indicates that Raman microscopy is capable to monitor hepatocyte-specific CYP induction without labeling.

**Raman microscopy enables quantitative analysis of CYP activity.** To further investigate the correlation between Raman shifts assigned to CYPs (1370 cm$^{-1}$ and 1636 cm$^{-1}$) and CYP activity, we examined the inducer concentration-dependent, inducer type-dependent, down-regulatory, and time-dependent effects on CYP activity by Raman microscopy at the single-cell level. As the shape of the peak at 1370 cm$^{-1}$ was strongly affected by the adjacent 1360 cm$^{-1}$ peak assigned to reduced hemes, we chose the Raman shift at 1636 cm$^{-1}$ for the quantitative analysis of CYP activity. The signal processing method used for the analysis is shown in Supplementary Fig. 5.

First, we applied different concentrations of RIF in a gradient to the cell culture. The high spatial resolution of Raman imaging allowed us to investigate the cell-cell differences in CYP3A4 activity, as shown in the Raman images (Fig. 3a–d). Both the Raman intensity at 1636 cm$^{-1}$ and the luminescence intensities increased with increasing RIF concentration (Fig. 3e). The higher background in the Raman results could be a signal from oxidized cyt $c$ or other types of CYPs. The Raman intensity at 600 cm$^{-1}$ indicated that cyt $c$ was constant regardless of CYP induction (Supplementary Fig. 6a), which was supported by the western blotting results (Fig. 1d), indicating that the increase in the 1636 cm$^{-1}$ was mainly caused by the induction of CYPs. The analysis of the Raman intensity at the single-cell level is shown in Fig. 3i (boxplot). The ratio above the third quantile of the control condition gradually increased from 25 to 76% when the RIF concentration reached 4 μM, and the ratio below the first quantile compared with control conditions decreased from 25 to 8%

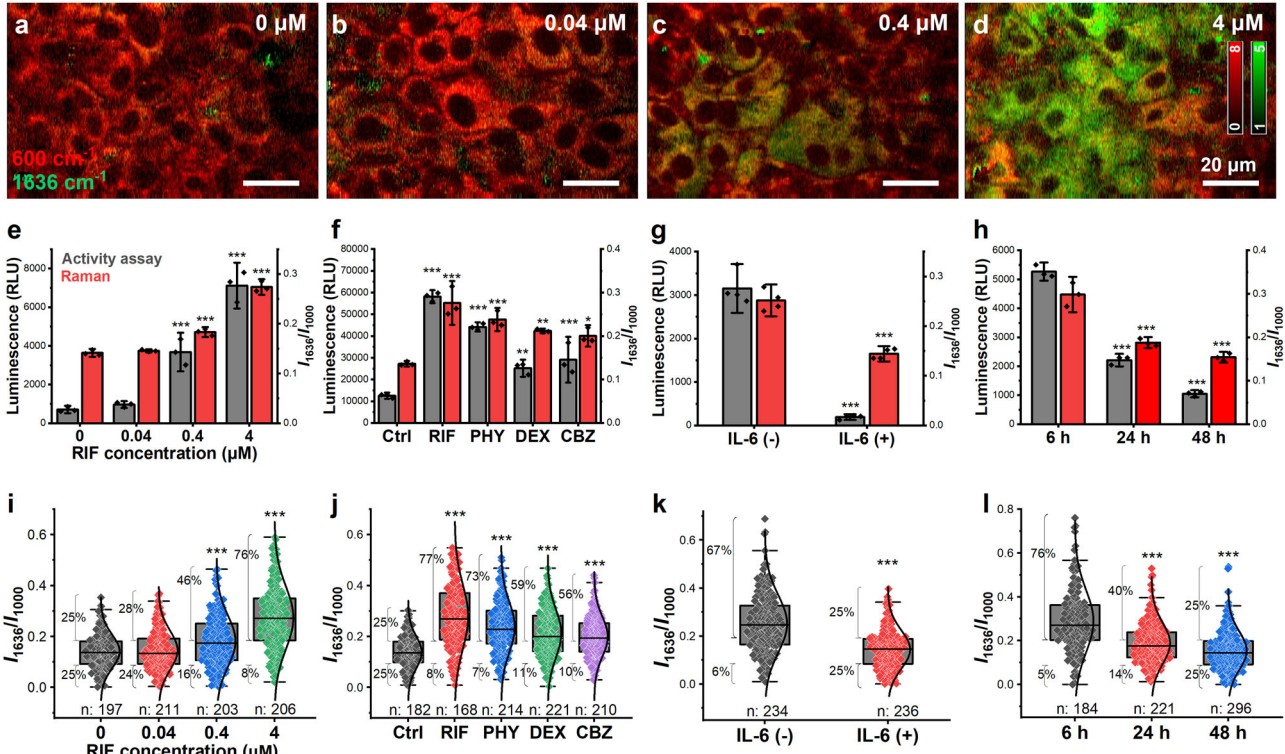

**Fig. 3 CYP activity at various conditions measured by Raman microscopy. a–d** Reconstructed Raman images at different RIF concentrations from 0, 0.04, and 0.4 to 4 µM. Images constructed at 1636 cm$^{-1}$ are labeled in green, and images constructed at 600 cm$^{-1}$ are labeled in red. The number of 1636 cm$^{-1}$ positive cells increased with increasing concentration of RIF. Scale bars, 20 µm. **e–h** Comparison of CYP activity measured with luminescence assay and average Raman intensity at 1636 cm$^{-1}$ with RIF in a gradient (**e**), various CYP3A4 inducers (**f**), IL-6 (**g**), and at the different culture time (**h**), of which the Raman intensity of individual cells is shown in (**i–l**), respectively. Error bars **e–h** indicate SD between triplicates. Box plots illustrate the spread within cell populations (center line, median; box, quartiles; whiskers, 1.5× interquartile range). Statistical significance was evaluated by one-way ANOVA followed by post hoc Tukey-Kramer's range tests (compared with the left column/box). ***$P < 0.001$; **$P < 0.01$; *$P < 0.5$.

(Fig. 3i), demonstrating an increase in higher CYP-expressing populations. We also investigated the universal usage of Raman measurements on other CYP inducers (Fig. 3f, j), including phenytoin (PHY), dexamethasone (DEX), and carbamazepine (CBZ), all of which showed the similar tendency as that seen in the luminescence assay of CYP3A4 (Fig. 3f). The induction of CYP1A2, CYP2B6, and CYP2C9 is shown in Supplementary Fig. 7, indicating that the change in CYP3A4 is dominant.

Inflammation and infection downregulate the expression and activity of CYPs, altering the pharmacokinetics and clearance of drugs[47]. The inhibitory effect varies among patients due to pharmacogenetic variations in the inflammatory pathway[48]. Therefore, we investigated the possibility of using Raman microscopy to analyze the downregulation of CYP by interleukin-6 (IL-6), a principal regulator of the hepatic acute-phase response[49]. 2 ng/mL IL-6 was added to HepaRG cells and treated for 48 h to suppress CYP3A4 expression. Downregulation of CYP3A4 expression was confirmed by measuring CYP3A4 activity (Fig. 3g). The Raman intensity at 1636 cm$^{-1}$ decreased, consistent with the results of CYP3A4 activity assay (Fig. 3g, k).

Next, we evaluated the dynamics of CYP3A4 expression without any external interference. HepaRG cells expressed higher levels of CYP3A4 in the first several hours after thawing, which decreased gradually until 48 h. This natural decrease was confirmed by analyzing the intensity of the peak at 1636 cm$^{-1}$ (Fig. 3h, l).

To eliminate the possible contamination of Raman scattering from inducers or down-regulators. The Raman spectra of the stock solution of each chemical are shown in Supplementary Fig. 8. No overlapping at 1370 cm$^{-1}$ and 1636 cm$^{-1}$, as well as no

characteristic peaks of these inducers and down-regulators, were found in the cell spectrum.

Taken together, the analysis of the inducer concentration-dependent, inducer-type-dependent, down-regulatory, and time-dependent effects demonstrated that 1636 cm$^{-1}$ can be utilized as a Raman indicator to detect CYP activity under various conditions. This suggests that Raman microscopy can be applied as a screening method to evaluate the modulation of CYP activity by drugs and inflammatory stimuli.

**Mechanism-based inhibitor binds to active CYP**. To prove the feasibility of using the 1636 cm$^{-1}$ Raman shift to indicate induced CYP, a mechanism-based inhibitor was applied to the cell culture. Mechanism-based inhibitors bind to CYP as a substrate analog and trigger the catalysis reaction[50]. The catalysis intermediate binds to the heme core or apoprotein and blocks the entrance of other substrates to irreversibly inactivate the enzyme. Azamulin (AZA), a highly specific and mechanism-based inhibitor of CYP3A4[51], can bind to the heme core of CYP3A4 and cause ferric LS to high-spin (HS) transition[52] (Fig. 4a). The Raman spectra of HepaRG cells with or without AZA treatment were collected (Fig. 4b), and the CYP3A4 activity assay was performed to confirm the inhibition effect, as shown in Fig. 4c. Both the Raman signal at 1636 cm$^{-1}$ and 1370 cm$^{-1}$ decreased after 10 min of treatment with 10 µM AZA (Fig. 4c). The decrease in the Raman shift was smaller than that in the activity assay, presumably due to the signal of oxidized cyt $c$ or other CYP subtypes at the same wavelength, for reasons discussed in the following paragraph.

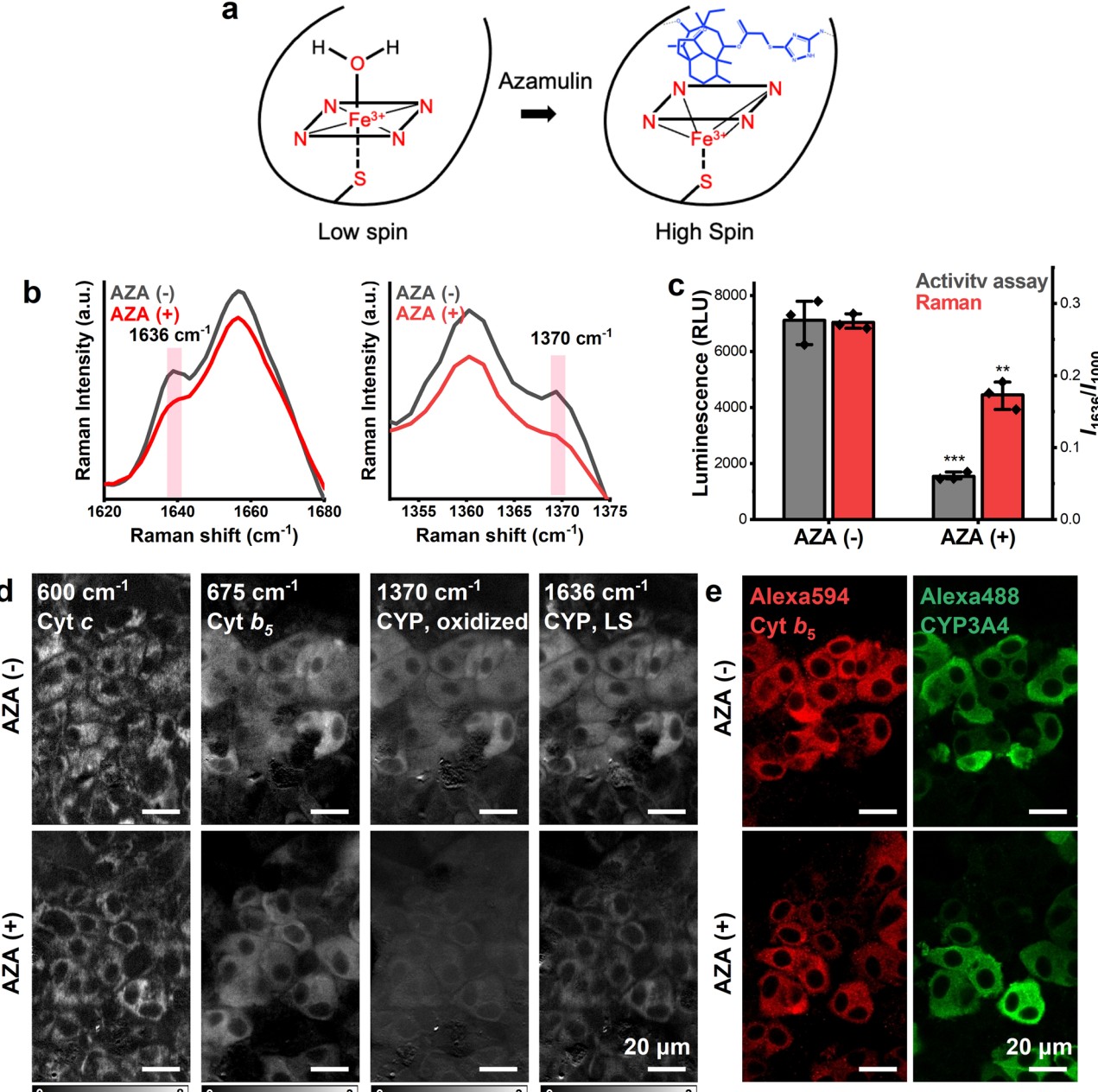

**Fig. 4 Detection of CYP inhibition after azamulin (AZA) treatment. a** Schematic diagram of a possible mechanism of AZA binding with CYP. Active CYPs maintain a ferric low-spin (LS) state by using water as a ligand. After AZA enters the enzymatic pocket, water is replaced by AZA, which causes LS to high-spin (HS) transition. Although metabolic intermediates of AZA were not identified, the crystal structure indicated the pleuromutilin head is required for mechanism-based inhibition. The possible ligation was depicted as a black dotted line. **b** The Raman shifts at 1370 cm$^{-1}$ and 1636 cm$^{-1}$ decreased after treatment with AZA. **c** Comparison of average Raman signal at 1636 cm$^{-1}$ and CYP3A4 activity assay. Error bars indicate SD between triplicates. ***$P < 0.001$; **$P < 0.01$. **d** Reconstructed Raman images at 600 cm$^{-1}$, 675 cm$^{-1}$, 1370 cm$^{-1}$, and 1636 cm$^{-1}$. Scale bars, 20 μm. **e** Immunostaining of cyt $b_5$ and CYP3A4 at the same positions of Raman measurement. Scale bars, 20 μm.

The sensitivity of Raman spectroscopy to the spin-state transition allowed us to evaluate the inhibition of CYP activity in living cells using the Raman peak at 1636 cm$^{-1}$, as shown in Fig. 3d. A decrease of cytoplasm contrast was found in the Raman images of 1370 cm$^{-1}$ and 1636 cm$^{-1}$. Furthermore, immunofluorescence staining (Fig. 4e) and western blotting (Supplementary Fig. 9) of CYP3A4 demonstrated that AZA treatment did not alter the amount of CYP3A4, suggesting that the decrease in Raman intensity at 1636 cm$^{-1}$ was associated with the change in the activity of CYP but not the amount. In the previous studies using recombinant protein, AZA bound to ferric CYPs and caused a

ferric LS to HS transition[52]. The spin-state transition of CYP protein upon substrate binding has been investigated using resonance Raman spectroscopy due to its high sensitivity and possibility to measure under physiological conditions. Three spin-state marker bands are sensitive to the changes in porphyrin core size caused by the transition of spin state. These bands were found around 1500 cm$^{-1}$ (ν3), 1590 cm$^{-1}$ (ν2), and 1640 cm$^{-1}$ (ν10) for the LS state and around 1490 cm$^{-1}$, 1570 cm$^{-1}$, and 1620 cm$^{-1}$ for the HS state[28,30,34]. In our measurement, 1500 cm$^{-1}$ and 1590 cm$^{-1}$ were not detectable due to the interference of other cellular components or the possible lower efficiency of Raman

scattering at 532 nm. A decrease of $1636\,cm^{-1}$ was observed which allowed us to evaluate inhibition in living cells using this peak. Although $1370\,cm^{-1}$ is not a so-called spin-state marker, the decrease in intensity was observed in our measurement (Fig. 4b), which might be associated with the reduction of CYP after binding to AZA.

We also performed the inhibition experiment using CYP3A4 overexpressed microsome, which is a simplified subcellular system preserving CYP activity. Microsomes were treated with $10\,\mu M$ AZA for 10 min, and the Raman spectra were collected (Supplementary Fig. 10). We detected the transition from $1636\,cm^{-1}$ to $1621\,cm^{-1}$ and even shift from $1370\,cm^{-1}$ to $1367\,cm^{-1}$ as well as the decrease in intensity of $1636\,cm^{-1}$ and $1370\,cm^{-1}$. In living cell condition, Raman shifts $\sim 1621\,cm^{-1}$ were overwhelmed by other cellular peaks, and $1370\,cm^{-1}$ was affected by the adjacent peak at $1360\,cm^{-1}$. Therefore, the decrease in $1636\,cm^{-1}$ was used to evaluate CYP inhibition in living cells.

From the spectroscopic view, the decrease in $1636\,cm^{-1}$ without any change in the protein amount corresponds to the fact that the inhibition by AZA is due to the spin-state transition. After treatment with AZA, we found that the Raman image of $1636\,cm^{-1}$ provided a mitochondrial contrast, which was also observed with reduced cyt $c$ ($600\,cm^{-1}$). Since AZA does not affect the redox state of cyt $c$ in HeLa cells (Supplementary Fig. 11) and the amount of cyt $c$ in HepaRG cells (Supplementary Fig. 6), these results indicate that the $1636\,cm^{-1}$ Raman signal also contains oxidized cyt $c$, contributing as a constant background in the measurements. The alternations of oxidized CYP signals induced the change in the Raman signal shown in Fig. 3 and Fig. 4.

**CYP induction with glycogen depletion**. The liver is an essential organ associated with not only drug metabolism but also glucose metabolism. The crosstalk between drug and glucose metabolism has been investigated to understand drug-induced metabolic disorders[53]. Nuclear receptors for drugs[54], including pregnane X receptor (PXR), constitutive androstane receptor, retinoid X receptor, have been shown to modulate enzyme expression in glucose metabolism[55]. In particular, the induction of CYP3A4 by RIF is mediated by the activation of PXR[33].

We investigated the intracellular glycogen storage and RIF-induced CYP activity in situ using Raman microscopy. HepaRG cells resemble the features of adult hepatocytes with the ability to express CYP3A4 ($1636\,cm^{-1}$), cyt $b_5$ ($675\,cm^{-1}$), and store glycogen ($940\,cm^{-1}$), and lipids ($1450\,cm^{-1}$). As each of these hepatic functional biomolecules features in specific Raman shifts, the time-dependent changes of these molecules were all visualized by the Raman microscopy. The time-course cell Raman images at different culture days and conditions are shown in Fig. 5a. The increase of cellular glycogen with the culture time was imaged using the reconstructed Raman images at $940\,cm^{-1}$.

Interestingly, we found that the amount of glycogen decreased as CYP expression increased (Fig. 5b). Figure 5c shows the negative correlation between CYP induction and glycogen storage, where each spot demonstrates a single cell. More high-CYP and low-glycogen cell populations were found after treatment with RIF (Fig. 5c). The inhibitory effect of CYP inducers on glycogen storage was detected when treated with other inducers (Supplementary Fig. 12). Following this, we confirmed the inhibitory effect of CYP inducers on glycogen storage by PAS staining (Fig. 5d). Our findings are in accordance with in vivo studies in which RIF treatment was found to reduce the glycogen content in rat livers[56]. The possible mechanism is through the activation of PXR which promotes the expression of

PECPK1 and G6P30, two important enzymes in glucogenesis and glycogenolysis[57].

Meanwhile, no significant difference was found in the Raman intensity at $600\,cm^{-1}$ (cyt $c$), $675\,cm^{-1}$ (cyt $b_5$), and $1450\,cm^{-1}$ (lipid) (Supplementary Fig. 13) after CYP induction, indicating a potential lack of the side effects of RIF on mitochondrial function and lipid metabolism. With a single scan of the cell sample, the negative regulation of glycogen storage was observed simultaneously with CYP induction by Raman microscopy, demonstrating that the multiplexity of drug response can be assessed in a non-destructive manner.

## Discussion

This research proposed a label-free and high-content method to assess CYP activity in living hepatocytes by Raman microscopy. The appearance and intensity of the Raman peaks provided specific patterns characteristic of hepatocytes under drug administration. CYP activity was visualized and quantified by characteristic peaks of heme at $1370\,cm^{-1}$ and $1636\,cm^{-1}$ since active CYPs contain an oxidized iron atom at LS state in the central catalysis pocket. Meanwhile, a negative regulation of glycogen storage, visualized at $940\,cm^{-1}$, was observed together with CYP induction, demonstrating the potential of drug toxicity towards glucose metabolism. Thus, without any external labeling, we were able to evaluate hepatocyte-specific modulation of CYP activity and the associated effect on glucose metabolism, providing an information-rich analysis of drug metabolism without disturbing cell homeostasis.

Induction of CYPs in living hepatocytes was visualized and characterized as oxidized and LS form in living hepatocytes by resonance Raman microscopy. Our approach provides a platform to understand the drug and CYP interaction in living cells by accessing the catalytic core of the enzyme. We demonstrated that induced CYPs in living cells are in the ferric state and LS state. Label-free visualization of active CYPs paves a direct way to measure CYP activity in living hepatocytes. Although it is difficult to identify the different types of CYPs, Raman microscopy can be utilized as a pre-screening method for total CYP induction and inhibition prior to the detailed investigation of the specific sub-types, which can accelerate the process of drug screening.

We found that resonance Raman microscopy can simultaneously observe the effect of the drug on CYP dynamics and glycogen storage without any labels in live cells. An increase in blood glucose in humans after RIF administration was firstly reported in 1982[58], almost 10 years after FDA approval. Our results suggest that RIF-induced hyperglycemia could be through a mechanism of glycogen breakdown. Other CYP inducers utilized in this research have also been reported to induce hyperglycemia[59–61] and should therefore be used with caution when administrated to diabetes patients. Therefore, we provided a simple and non-destructive method to screen for CYP induction and adverse effects on glucose metabolism, which will predict drug side effects more comprehensively and facilitate the prescription of drugs for diabetic patients. Our findings suggest that Raman microscopy has great potential for assessing and predicting multiple drug toxicity, which remains a challenge in new drug development[62].

The high-content visualization of hepatic functional molecules (CYPs, glycogen, and cyt $b_5$), as well as general cell markers (cyt $c$, protein, and lipids), allows for a phenotypic and unbiased evaluation on the hepatocyte functions and maturation levels. Quality control of regenerative cell products prior to transplantation is essential to ensure the application from the bench to the bedside. Label-free and non-destructive Raman microscopy will also be a promising analytical tool for quality control and clinical translation of stem cell differentiated hepatocytes.

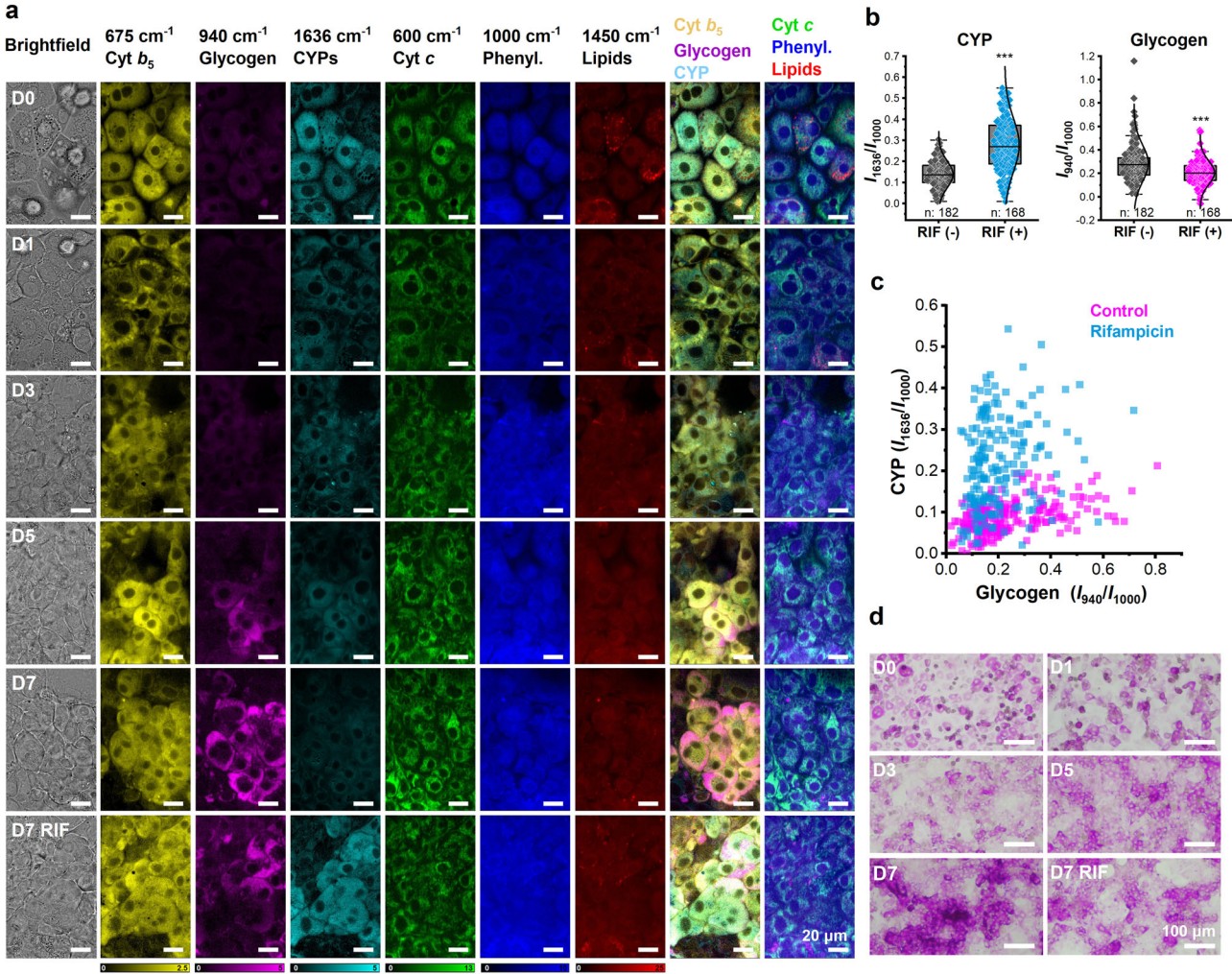

**Fig. 5 CYP induction with glycogen depletion. a** Hyperspectral Raman images reveal dynamic changes in hepatic functional molecules (cyt $b_5$, glycogen, and CYPs) and general cellular components (cyt $c$, phenylalanine, and lipids) from D0 to D7. An accumulation of glycogen is observed, whereas treatment of CYP inducer deceases the glycogen amount. Scale bars, 20 μm. **b** Quantitative analysis of the Raman intensity at 940 cm$^{-1}$ and 1636 cm$^{-1}$ shows decreased glycogen with increased CYP after CYP inducer treatment. Box plots illustrate the spread within cell populations (center line, median; box, quartiles; whiskers, 1.5× interquartile range). ***$P < 0.001$. **c** Single-cell-based dot plot with normalized Raman intensity at 940 cm$^{-1}$ ($x$ axis) and 1636 cm$^{-1}$ ($y$ axis). Each dot indicates one cell. **d** PAS staining demonstrates the accumulation of glycogen during the 7-day culture and consumption with RIF treatment. Scale bars, 100 μm.

In conclusion, this study provides a non-destructive and high-content strategy for the evaluation of CYP activity without the need for any external labels. The label-free feature of Raman microscopy enables a multiplex and comprehensive evaluation of drug metabolism and glucose metabolism at the single-cell level, providing a new way for future drug development schemes.

## Methods

**Cell culture**. Cryopreserved differentiated HepaRG cells (HPR116), basal hepatic cell medium (MIL600), HepaRG thawing/plating/General purpose medium supplement (ADD670), HepaRG induction medium supplement (ADD650), and HepaRG culture medium supplement (ADD620) were purchased from Biopredic International. Each supplement was added to the basal hepatic cell medium to obtain the HepaRG thawing/plating/general-purpose, induction, and culture media, respectively. HepaRG cells were thawed and cultured in 35-mm dishes (SF-S-D12; Fine Plus International) with quartz substrate (ø 12 mm) affixed to the bottom to enable Raman observation. For the seeding of cells on the quartz substrate, cloning rings with an outer diameter of 10 mm and an inner diameter of 8 mm were placed on the quartz, and $1.2 \times 10^5$ HepaRG cells were seeded into each cloning ring. HepaRG cells were allowed to attach to quartz substrates for 24 h, at which time the cloning rings were removed, and fresh HepaRG thawing/plating/general purpose medium was added. The HepaRG cells were continuously maintained in HepaRG thawing/plating/general purpose medium 5 days at 37 °C with

5% $CO_2$. The induction of CYP3A4 activity was conducted for 48 h using an induction medium containing 4 μM RIF (189-01001), 100 μM PHY (166-12082), 100 μM DEX (047-18863), or 100 μM CBZ (034-23701) purchased from Fujifilm Wako. Alternatively, the inhibition of CYP3A4 activity by 10 μM AZA (18748; Cayman) was performed after 48 h induction of CYP3A4 by RIF.

Cryopreserved primary human hepatocytes (PHHs, Lot No. HC2-50), OptiThaw hepatocyte isolation kit (K8000), OptiCulture media kit (K8300M), and OptiPlate hepatocyte media (K8200) were purchased from Sekisui XenoTech. PHHs were thawed using OptiThaw hepatocyte isolation kit and cultured in 35-mm dishes with quartz bottom (SF-S-D12; Fine Plus International) to enable Raman observation. For the seeding of PHHs on the quartz substrate, silicone rings with an inner diameter of 15 mm were placed on the quartz pre-coated with collagen, and $3.2 \times 10^5$ PHHs were seeded into each ring. After cells adhered to the culture surface, Raman measurement was performed. HepaRG cells with the same density were taken as controls.

**CYP activity measurement**. CYP1A2 (V8422), CYP2B6 (V8322), CYP2C9 (V8792), and CYP3A4 (V9002) activity were measured using P450-Glo$^{TM}$ assay purchased from Promega. The measurement of each CYP activity was conducted according to the manufacturer's protocol. After treatment with inducers or inhibitors, the culture medium was replaced with a fresh medium containing luciferin substrates and incubated at 37 °C with 5% $CO_2$. After incubation, 50 μL of substrate medium was aspirated and transferred to a 96-well white flat-bottom plate (Corning). Luciferin detection reagent (50 μL) was added to each well to initiate a luminescent reaction. The resulting luminescence was measured using a microplate reader (Synergy HTX; BioTek).

**Immunofluorescence staining**. Immediately after the Raman measurements, the HepaRG cells were fixed with 4% paraformaldehyde for 20 min at room temperature (RT) and stored at 4 °C until further analysis. Cells were permeabilized with 0.1% Triton X-100 in PBS for 10 min and blocked with 4% bovine serum albumin (A2153; Sigma-Aldrich) for 1 h at RT. A primary antibody working solution containing mouse anti-CYP3A4 (1:1000; SAB5300118; Sigma-Aldrich) and rabbit anti-human cyt $b_5$ (1:500; ab69801; Abcam) was prepared in 1% bovine serum albumin solution, applied to the cell samples, and incubated overnight at 4 °C. The following day, the samples were washed three times with PBS and then immersed in a secondary antibody solution containing Alexa Fluor 488 goat anti-mouse antibody (10 µg/mL; A11001; Invitrogen) and Alexa Fluor 594 goat anti-rabbit antibody (10 µg/mL; A11012; Invitrogen) for 1 h at RT. After washing with PBS three times, the cells were counterstained with 1 µM DAPI solution (D1306; Invitrogen) at RT. The cells were then washed twice with PBS, and the dishes were stored and protected from light at 4 °C until image acquisition. Fluorescent images were captured using a confocal laser scanning microscope (A1; Nikon).

**Western blotting**. After drug treatment, the HepaRG cells were washed twice with cold PBS and lysed on ice for 30 min using RIPA buffer containing a protease inhibitor cocktail (1:100; P8340; Sigma-Aldrich). The lysate was collected using a cell scraper and centrifuged at $12,000 \times g$ for 20 min at 4 °C to remove cell debris. The protein amount was measured using a BCA protein assay kit (23227; Thermo Fisher Scientific). Proteins were electrophoresed using 12% sodium dodecyl sulfate-polyacrylamide gels (4568043; Bio-Rad) and transferred to a low-fluorescent polyvinylidene fluoride membrane. After blocking with 5% ECL blocking agent (RPN2125; GE Healthcare) in TBST (0.1% Tween-20 in TBS) for 1 h at RT on a shaker and rinsing with TBST twice, the membranes were separately immersed in mouse anti-CYP3A4 (1:2000, Sigma-Aldrich), rabbit anti-cyt $b_5$ (1:1000, Abcam), mouse anti-cyt $c$ (1:200, Abcam, ab65311), and rabbit anti-β-actin (1:1000; 4970 S; CST) primary antibody overnight. The following day, horseradish peroxidase-conjugated anti-mouse (NA931VS; Bio-Rad) or anti-rabbit (NA934VS; Bio-Rad) secondary antibodies (1:10,000) were applied for 1 h at RT, followed by washing three times in TBST buffer. The protein bands were visualized using an enhanced chemiluminescence detection system (RPN2232; GE Healthcare) and detected using the ChemiDOC MP imaging system (Bio-Rad).

**Slit-scanning Raman microscope**. All the Raman hyperspectral datasets were obtained using a home-built slit-scanning confocal Raman microscope equipped with a 532 nm CW laser (millennia eV; Spectra-Physics) with a power density of 3 mW/$\mu m^2$ [23,24]. The line-shaped laser light was produced using cylindrical lenses and focused on the sample using a ×40 water immersion objective lens (NA 1.25 CFI Apochromat Lambda S; Nikon). The Raman scattered light was collected with the same objective lens and passed into a spectrograph (MK-300; Bunkoukeiki) through a long-pass edge filter (LP03-532RU-25; Semrock). The axial resolution similar to the confocal effect was achieved by the slit placed in front of the spectrometer[63]. The light dispersed by the grating (1200 L/mm) was then detected using a cooled CCD camera (PIXIS 400 B; Teledyne Princeton Instruments) with an exposure time of 5 s. The spectral resolution was 1.5 cm$^{-1}$. 400 spectra in a line were collected with one exposure. Images were acquired by scanning with a single-axis galvanometer mirror. Each high-resolution Raman image contains 400 pixels × 252 pixels for a total of 100,800 spectra per image (Figs. 1, 2, 4, and 5). For quantitative analysis, each image contains 400 pixels × 84 pixels, total 33,600 spectra per image (Fig. 3). With the hyperspectral Raman imaging implemented by our slit-scanning Raman microscope, we can provide a large-scale analysis at the single-cell level.

**Data processing**. Raman hyperspectral images were processed using the following procedures. Cosmic rays were removed by the median filter and singular value decomposition (SVD) was applied to reduce noise and loading vectors contributing to the image contrast were chosen. Following SVD processing, a baseline correction was conducted at individual Raman shifts by approximating and subtracting the background signal with a straight line to avoid interference from adjacent Raman peaks (Supplementary Fig. 5). We used peak height instead of the area under the curve for quantification of Raman intensity to reduce possible contaminations of other peaks at the same area. The intensity distribution of the designated Raman shift was mapped to show as Raman images.

The procedure of the single cell-based quantification of Raman intensity is illustrated in Supplementary Fig. 5. Reconstructed Raman images at 750 cm$^{-1}$ (all cytochromes), 940 cm$^{-1}$ (glycogen), and 1000 cm$^{-1}$ (phenylalanine) were merged to present clear cell boundaries and the shape of the nuclei. The manual segmentation of the cytoplasm was performed by subtracting the area of the nucleus from the whole cell body. Individual cell masks were then created and applied to the spectrum data without SVD processing for acquiring the average spectrum of each cell used for the following quantitative calculations. The Raman intensity of the target shifts was calculated after baseline correction to reduce any interference from adjacent Raman shifts. Data points that did not provide a sufficient signal-to-noise ratio were removed. To compare the Raman intensities of the samples under various conditions, a Raman intensity of 1000 cm$^{-1}$ (phenylalanine, a typical Raman marker of protein) was used as an internal reference for the normalization of the designated Raman shifts[64].

**Raman measurement of microsome**. Supersomes overexpressed human CYP3A4 (456207) were purchased from Corning. 10 µM AZA was added into supersomes solution which contains 2 µM CYP3A4. After being treated with 10 µM AZA for 10 min, Raman observation was conducted. Supersomes treated with DMSO were taken as controls.

**Statistics and reproducibility**. The quantitative data presented in this study were obtained from three independent cultures. Statistical analysis was firstly assessed with a one-way analysis of variance, and individual differences were tested using post hoc Tukey–Kramer's range tests according to unequal sample sizes. All calculations were performed using OriginPro 2021 (OriginLab Corporation, MA, USA). Statistical significance was set as ***$P < 0.001$; **$P < 0.01$; *$P < 0.05$; n.s. (not significant): $P \geq 0.05$.

**Reporting summary**. Further information on research design is available in the Nature Research Reporting Summary linked to this article.

## Data availability

All data necessary to evaluate our conclusions are included in the main manuscript and the supplementary material. All data used in this paper are available from the corresponding author on reasonable request. Uncropped western blot images are provided in Supplementary Fig. 14.

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

## Acknowledgements

We thank K. Yamamoto for her help with experiments and analyses. This research was supported by AIST-Osaka University Advanced Photonics and Biosensing Open Innovation Laboratory (PhotoBio-OIL). This study was funded by JST-CREST and JST COI-NEXT under Grant Numbers JPMJCR1925 and JPMJPF2009.

## Author contributions

S.F. and K.F. conceived the concept; M.L., Y.N., S.I., Y.K., S.F., and K.F. designed the experiments; M.L. and Y.N. performed the experiments; M.L., Y.N., and K.F. analyzed the data; M.L., Y.N., and K.F. wrote the manuscript with input from S.I., Y.K., S.F. All authors have read and agreed to the published version of the manuscript.

## Competing interests

The authors are the inventors of a patent on Raman measurement of CYP activity submitted by AIST and Osaka University.
