## [Peer Review File · Communications Biology]

Reviewers' comments:

Reviewer #1 (Remarks to the Author):

Authors applied custom-built slit scan Raman microscope to develop non-destructive and high-content method for visualizing and quantifying intracellular CYP activity under drug administration.

Induced CYP by RIF in living hepatocytes is visualized and quantified.

The Raman spectra of HepaRG cells with/without azamulin treatment were observed, and the CYP activity assay was performed to confirm the inhibition effect.

Finally, the intracellular glycogen storage and the inhibitory effect of CYP inducers on glycogen storage was observed and confirmed by PAS staining.

Overall, this is a well-written manuscript with clear experimental results.

This work will bring many interesting future applications.

Before publication, I want authors to address below minor comments.

Current experiments are in-vitro experiments. It will be helpful to explain how this method can be applied to in-vivo. Especially, most Raman signals are resonant signal by 532nm excitation which is difficult to translate to in-vivo measurement.

In Fig S5, authors used peak height instead of area under the curve (AUC). Is there any specific reason for that?

What is the axial resolution of the microscope? Cultured cells for few days may not be monolayered. Depending on the resolution, signals from multiple cells may be integrated.

Reviewer #2 (Remarks to the Author):

This paper investigated the potential to use Raman spectra to assess CYP450 activities in alive cells. The authors applied Rifampicin to induce CYP3A4 expression in HepaRG cells and found the Raman intensity at 1636 cm^{-1} increased with RIF. Meanwhile, glycogen was down regulated as CYP expression increased, which was attributed to the activation of PXR which promotes glucogenesis and glycogenolysis. Other CYP inducers were employed to confirm RIF results. The study is interesting and well designed; however, significant improvement is needed before it could be accepted for publication. My detailed comments are as follows:

Major comments:

1. CYPs are highly expressed in primary hepatocytes. Although HepRG is a widely used hepatocyte-like model, considering the fact that it is a mixture of hepatocytes-like and biliary-like cells, primary hepatocytes (human or rat) should be used as positive control.
2. In Fig. S8, Raman spectrum of cyt c solution treated with azamulin was employed to an conclusion that azamulin didn't influence cyt c Raman shift. However, the reviewer found it is hard to draw this conclusion since the baseline shift made it hard to judge the Raman shift. Besides, the reviewer is not sure if this study could reflect real situation in cells.
3. In Fig. S9. Raman images about the content of glycogen and CYP under treatment with various inducers were listed. However, there was no WB and function assay results to confirm if these inducers changed CYP protein expressions or functions.
4. It is still not clear if Raman shift at 1636 cm^{-1} only influence CYP3A4 or the combination of multiple CYPs. It would be great if the authors could investigate the results of other CYP450s (e.g., the induction/inhibition of CYP1A2).
5. The inducer utilized in this paper may influence multiple CYPs. RIF could not only induce the CYP3A4 expression, but also CYP2C9 and so on. Other CYP3A4 inducers in this paper may influence multiple CYPs as well. For example, Phenytoin is a known inducer of drugs metabolized by CYP2C, CYP2D, and CYP3A.
6. In the quantitative comparison, the author adopted $I_{\text{xcm-1}}/I_{1100\text{cm-1}}$ (As shown in the

figure). But what is the rationale that the peak intensity at 1000 cm^{-1} to be constant?

7. Fig.3. speculated that CYP inhibition after azamulin treatment led to the Raman shifts observed in HepRG cells. The hypothesis is, active CYPs maintain a ferric low-spin state by using water as a ligand. After azamulin enters the enzymatic pocket, water is replaced by azamulin, which causes low-spin to high-spin transition. It would be great if the author could use sub-cellular system to further approve their hypothesis, for example, CYP3A4 overexpressed microsomes/proteins, which should be commercially available.

Minor comments:

1. In the HepRG images (Fig. S2), in order to identify hepatocytes and biliary cells, the related biomarkers (by immunofluorescence staining) should be used to characterize the two kind of cells, not only use Raman spectra.

2. Raman spectroscopic characterization of hepatocytes and biliary cells in HepaRG culture showed significant differences (Fig. S2). The CYP inducer/inhibitor impacts on the biliary cells Raman signals are critical to evaluate the non-invasive method. It would be great if the author could compare the spectra in both cells instead of just hepatocyte-like cells. Therefore, Fig. S4 should be moved to the draft instead of in the supplemental part.

3. In general, CYP induction would be conducted by 72h inducing. Please list the rationale why only 48h was used in this study.

4. For CYP inhibition, why it was conducted after 48h induction? In general, it should be conducted in the cells without inducing treatment.

5. In this paper, each high-resolution Raman image contains 400 pixels \times 252 pixels for a total of 100, 800 spectra per image. What is the integration time for each spectrum and how long it may take to obtain such a high-resolution image?

6. The hyperspectral Raman images in this paper should provide their brightness scale.

Reviewer #3 (Remarks to the Author):

The authors of the manuscript "Label-free chemical imaging of cytochrome P450 activity by Raman microscopy" present studies in which they use Raman microscopy to visualize and evaluate the activity of cytochrome P450, more specifically CYP3A4, under the physiological condition in human hepatocytes. CYP3A4 is a crucial cytochrome P450 in human physiology since it is responsible for the metabolism of almost 50% of human drugs. Raman microscopy is a noninvasive, label-free technique that has great potential for studying critical physiological processes at the cellular level. The authors convincingly argue that the CYP distribution within the cells can be successfully monitored using Raman microscopy, and even more significantly, allow probing some of their activity. The authors support their spectroscopic findings with a broad array of other techniques, such as activity measurements, immunofluorescence staining, Western blotting, and statistical analysis. As such, I think the presented overall findings are of great importance and will be of great interest to the broad scientific community. However, while the general conclusions are correct, meaning that the authors present convincing arguments that they can monitor CYP in the cells and utilize the Raman microscopy to detect CYP activity under various conditions, there are several major problems, in my opinion, with data interpretation, that needs to be addressed before publication in Communications Biology.

Major problems:

1. The authors' interpretation of Raman data in Figure 1.a that led to the statement: "...ferric CYPs were the functional form in living human hepatocytes" (page 5, lines 11-12) is, in my opinion, incorrect. I agree that the cell induction results in the increase of the 1370 cm^{-1} band (Figure 1.a), which indicates that the protein exists in a ferric state (Biochemistry, 1992, 31, 4384-4393). However, there is an additional band of equal intensity in this spectrum at around 1360 cm^{-1} , which is close to the 1345 cm^{-1} , characteristic of ferrous state (Biochemistry, 1992, 31, 4384-

4393), meaning that the protein exists as a mixture of ferric and ferrous states. Actually, the authors do recognize that this band is associated with the reduced form (page 7, line 5). Please also note that this band is dominant in the spectrum of control protein, indicating that the protein before induction is mainly in the ferrous state. I recommend that the authors rewrite this paragraph and focus on making the point that they are able to convincingly visualize CYP using a 1370 cm⁻¹ oxidation band without concluding that the ferric form is the only form of CYPs in living cells.

2. The paragraph describing mechanism-based inhibition needs major revision:

2.a The 1370 cm⁻¹ band is an oxidation state marker, not a spin state marker. The addition of azamulin does not change the oxidation state but the spin state. As such, including the 1370 cm⁻¹ band as an indicator of mechanism-based inhibition is incorrect. I do agree, though, that the 1636 cm⁻¹ mode can be indicative of spin state change associated with the binding of azamulin. The transition from 6 coordinated low spin (6cLS) to 5 coordinated high spin (5cHS) state, which is typically reflected by nu10 mode shifting from around 1640 cm⁻¹ (for 6cLS) to 1620 cm⁻¹ (for 5cHS) (Biochemistry, 1992, 31, 4384-4393) which can be observed in the spectrum as lowering of the 1636 cm⁻¹ mode. Please rewrite this part accordingly.

2.b Figure 3.a – The schematic drawing of azamulin as coordinated to the heme is implausible. The authors of reference 44 showed that the azamulin binds within the heme pocket and replaces the water cluster, which triggers a switch from 6cLS state to 5cHS state (see Figure 3, ref. 44). Coordination of azamulin to the heme iron (as the Figure 3.a suggests) would most likely result in different UV-Vis response and produces a 6cLS state with a Soret maximum around 420 nm, which is not observed (ref 44, Figure 3). Furthermore, the sentence on page 9, lines 4-5 is not accurate. Binding to the heme core (meaning porphyrin macrocycle or its peripheral groups) is only one of many ways of mechanism-based inhibition; others are binding to the heme iron or active site amino acid residues. Please rewrite the sentence.

Minor issues:

3. Page 3, line 14 – needs reference

4. Page 3, line 117-18 – sentence “The functions of catalytic ...” is unclear, I am not sure what the authors mean

5. Page 4, lines 15-16 – “... which were assigned to oxidized heme ...”, please be more specific, modes around 1370 cm⁻¹ are associated with nu4, which is the oxidation state of heme proteins; however modes at around 1636 cm⁻¹ are associated with vinyl stretching modes and/or nu10 mode which is a heme iron spin state marker (not oxidation state marker).

6. It is possible (and quite likely) that the inducers or downregulator exhibit their own Raman spectrum; how did the authors account for the possible contribution of the inducers/downregulator scattering to their data analysis (page 7 and 8)

7. What is the spectral resolution of reported Raman spectra?

I recommend publishing this work after a major revision.

Reviewer #1 (Remarks to the Author):

Authors applied custom-built slit scan Raman microscope to develop non-destructive and high-content method for visualizing and quantifying intracellular CYP activity under drug administration.

Induced CYP by RIF in living hepatocytes is visualized and quantified. The Raman spectra of HepaRG cells with/without azamulin treatment were observed, and the CYP activity assay was performed to confirm the inhibition effect. Finally, the intracellular glycogen storage and the inhibitory effect of CYP inducers on glycogen storage was observed and confirmed by PAS staining. Overall, this is a well-written manuscript with clear experimental results. This work will bring many interesting future applications.

Before publication, I want authors to address below minor comments.

- 1. *Current experiments are in-vitro experiments. It will be helpful to explain how this method can be applied to in-vivo. Especially, most Raman signals are resonant signal by 532nm excitation which is difficult to translate to in-vivo measurement.***

We fully agree with this comment and recognized the significance of this technique for *in vivo* studies. Our group and also collaborators have already applied this technique in characterizing pathological status of NASH/NAFLD tissues, cancer tissues and heart tissues. Studies on colon cancer tissue, skin, bone, and brain tissues using 532 nm excitation were also reported by other researchers and groups. However, the autofluorescence is a major concern of Raman observation at 532 nm, especially liver organ, which is a highly vascularized tissue. Pretreatment like perfusion with non-fluorescent buffer is necessary if we want to apply this technique to fresh tissues.

- 2. *In Fig S5, authors used peak height instead of area under the curve (AUC). Is there any specific reason for that?***

Area under the curve (AUC) is a general method to quantify Raman intensity. We have tried both peak height and AUC, the results are almost same. Since area covers more peaks, e.g. Oxidative cyt *c* is around 1630 cm^{-1} and CYP is around 1640 cm^{-1} , using AUC gave more background signal and the possible contaminations of other peaks. Therefore, we choose peak height for quantification.

To avoid possible concern from readers, we have added one sentence in the M&M to describe why we choose peak height instead of AUC. Please see changes from Line 28 on Page 20 to Line 1 on Page 21. The sentence is also attached below for easy checking.

“We used peak height instead of area under the curve for quantification of Raman intensity to reduce possible contaminations of other peaks at the same area.”

- 3. *What is the axial resolution of the microscope? Cultured cells for few days may not be monolayered. Depending on the resolution, signals from multiple cells may be integrated.***

The slit at the entrance of the spectrophotometer works similarly as a pinhole in confocal microscopy (Kawata et al., J. Opt. Soc. Am. A, 1991) providing the axial resolution in our Raman microscope. We experimentally measured the axial resolution of our system by obtaining an *x-z* image of a sample containing polystyrene (PS) bead with a diameter of 350 nm. *y-z* image of PS beads was obtained by scanning the objective lens for the *z* direction. The Raman observation condition was set the same as cell measurement (532 nm, 3 $\text{mW}/\mu\text{m}^2$, 5 s/line). The Raman image at 1003 cm^{-1} was shown in Fig. R1a. The intensity profile through the bead was shown in Fig. R1b as blue dots and curve fit them with a Gaussian using ImageJ. The average FWHM of three different bead images was calculated as $0.81 \pm 0.13 \mu\text{m}$. Therefore, although cells are not monolayered, the axial resolution of our system is smaller than 1 μm . So, the signal we integrated is within a single cell rather than multiple cells.

Thanks to this comment, we realized that we should emphasize that our Raman microscope has the axial resolution. Therefore, more descriptions were added in M&M and can be found in Lines 14-15 on Page 20.

“The axial resolution similar to the confocal effect was achieved by the slit placed in front of the spectrometer.”

Fig. R1. Cross-section image of 350 nm PS beads and intensity profile of beads at axial direction. **a** The reconstructed Raman image at 1003 cm^{-1} , assigned to PS. Scale bar, $2\text{ }\mu\text{m}$. **b** Axial (z direction) intensity profile of a PS bead with Gaussian fitting (red plot) applied to calculate the FWHM of the axial response.

Reviewer#2 (Remarks to the Author):

This paper investigated the potential to use Raman spectra to assess CYP450 activities in alive cells. The authors applied Rifampicin to induce CYP3A4 expression in HepaRG cells and found the Raman intensity at 1636 cm^{-1} increased with RIF. Meanwhile, glycogen was down regulated as CYP expression increased, which was attributed to the activation of PXR which promotes gluconeogenesis and glycogenolysis. Other CYP inducers were employed to confirm RIF results. The study is interesting and well designed; however, significant improvement is needed before it could be accepted for publication. My detailed comments are as follows:

Major comments:

- CYPs are highly expressed in primary hepatocytes. Although HepRG is a widely used hepatocyte-like model, considering the fact that it is a mixture of hepatocytes-like and biliary-like cells, primary hepatocytes (human or rat) should be used as positive control.***

We thank the reviewer for this comment. Using primary hepatocytes as a positive control can strengthen this paper and make broader interests. According to the reviewer's comment, we performed Raman measurement on human primary hepatocytes (PHH, XenoTech, Lot No. HC2-50) twice and detected the Raman peaks of CYP (1370 cm^{-1} and 1636 cm^{-1} , inserts of Figure R1) in PHH after plating. HepaRG cells after plating were also observed on the same day for comparison. Except for targeted CYP peaks, two peaks at 1157 cm^{-1} and 1512 cm^{-1} were detected (indicated by asterisks) in PHH but not in HepaRG cells. These two peaks can be assigned to carotenoids, most likely β -carotene, which is the most abundant carotenoids in the liver. This result also indicates the biochemical differences between primary hepatocytes and artificial model hepatocyte cell line, thus using PHH as a positive control was necessary. So we added this figure as Supplementary Fig. 3.

It is well known that PHH gradually lost CYP expression after plating, and our Raman observation results also showed this tendency. Supplementary Fig. 3b shows time-course Raman observations of PHH, performed at day 0, day 1, day 2, and day 3. A decreasing tendency of 1370 cm^{-1} and 1636 cm^{-1} were observed. The decreasing trend of CYP was confirmed by immunofluorescence imaging of CYP3A4 (Supplementary Fig. 3c), which is same as our observation using HepaRG. We added the following paragraph in Lines 12-22 on Page 5 about the above discussion.

“We also measured primary human hepatocyte (PHH) as a positive control. PHHs are widely used in drug metabolism and screening study due to high basal expression of CYPs.³⁶ Primary hepatocytes were observed right after cells were fully adhered to the collagen-coated surface. The Raman shifts at 1370 cm^{-1} and 1636 cm^{-1} were detected in PHH (Supplementary Fig. 3a). The time-course observation demonstrated the decreasing tendency of 1370 cm^{-1} and 1636 cm^{-1} after plating (Supplementary Fig. 3b), which is consistent with the immunostaining of CYP3A4 (Supplementary Fig. 3c). The expression of CYP in PHH rapidly decrease once after plating, this known feature of primary hepatocytes was monitored by Raman microscopy.³⁷ Besides 1370 cm^{-1}

and 1636 cm^{-1} , we also found dominant peaks at 1157 cm^{-1} and 1512 cm^{-1} (Supplementary Fig. 3a), which can be assigned to carotenoids.³⁸ Using Raman microscopy, we confirmed the same vibrational modes coming from CYPs in both PHH and HepaRG cells. On the other hand, the difference of these cell lines was also clarified, demonstrating the potential of label-free Raman on identifying features of different cell lines based on their intrinsic cellular components.”

Supplementary Fig. 3 Raman measurement of primary human hepatocytes (PHHs). **a** The Raman spectra of PHH (red trace) and HepaRG cells were taken after cells are fully adhered to the culture surface. The Raman peaks at 1370 cm^{-1} and 1636 cm^{-1} were enlarged and shown in the inserts. Asterisks indicate the Raman peaks (1157 cm^{-1} and 1512 cm^{-1}) detected in PHH but not in HepaRG cells, which can be assigned to carotenoids stored in liver. **b** The reconstructed Raman images at 1370 cm^{-1} and 1636 cm^{-1} , which can be assigned to oxidized and low-spin heme, respectively. A time-dependent decrease of Raman signals was observed. Scale bars, $20\text{ }\mu\text{m}$. **c** Immunofluorescence staining of CYP3A4 and nuclei counterstaining by DAPI. Scale bars, $50\text{ }\mu\text{m}$.

5. *In Fig. S8, Raman spectrum of cyt c solution treated with azamulin was employed to an conclusion that azamulin didn't influence cyt c Raman shift. However, the reviewer found it is hard to draw this conclusion since the baseline shift made it hard to judge the Raman shift. Besides, the reviewer is not sure if this study could reflect real situation in cells.*

We agree with the reviewer's comment that a protein solution may not reflect the real situation in living cells. Therefore, we have tried a different approach. HeLa cells, which do not express CYPs, were treated with $10\text{ }\mu\text{M}$ azamulin for 10 min. The average spectra with and without Azamulin treatment were compared

and shown in Figure R5. We found no apparent changes in the characteristic peaks assigned to reduced cyt *c* as well as the intensity. Although we are aware of the differences between HeLa cells and HepaRG cells, there is no particular reason that azamulin cannot reach the mitochondria in HeLa cells, and this approach is the most relevant way we could try. We believe that the cell-based approach is much closer to the real condition in HepaRG cells, at least better than simple protein solution. We added this result as Supplementary Fig. 11 and deleted the result using cyt *c* solution, which is not sufficient. Revisions in the manuscript can be found in Lines 8-9 on Page 13.

“Since AZA does not affect the redox state of cyt c in HeLa cells (Supplementary Fig. 11) and the amount of cyt c in HepaRG cells (Supplementary Fig. 6)”

Another direct evidence that azamulin works on CYPs is the microsome experiment that the reviewer suggested us to do. After azamulin treatment, decrease and shift of Raman peaks were detected.

Supplementary Fig. 11 Comparison of HeLa spectra with and without AZA treatment. HeLa cells treated with 10 μ M AZA for 10 min and Raman measurement was performed after the treatment. No significant difference was found in the characteristic peaks (asterisks) assigned to reduced cyt *c*.

6. *In Fig. S9. Raman images about the content of glycogen and CYP under treatment with various inducers were listed. However, there was no WB and function assay results to confirm if these inducers changed CYP protein expressions or functions.*

The result of CYP3A4 activity related to Fig. S9 has been shown in Fig. 2f (grey bars). We also further investigated on the induction effects of these inducers on CYP1A2, 2B6, and 2C9 and provided as Supplementary Fig. 7 and also attached below.

Supplementary Fig. 7 The effects on CYP activity after treatment with various inducers. **a** 1A2 activity. **b** 2B6 activity. **c** 2C9 activity. **d** 3A4 activity.

7. *It is still not clear if Raman shift at 1636 cm⁻¹ only influence CYP3A4 or the combination of multiple CYPs. It would be great if the authors could investigate the results of other CYP450s (e.g., the induction/inhibition of CYP1A2).*

We agree with the reviewer that our description on CYP3A4 or whole CYPs was ambiguous. Since Raman detects the vibrational modes of heme co-factor instead of amino acid sequence, theoretically, what we are observing is whole CYPs. To detect CYP signals, we treated HepaRG cells with RIF, which is a strong CYP3A4 inducer and CYP3A4 is a most abundant CYP subtype. Therefore, the amount of CYPs became large enough to be detected by Raman microscopy.

However, we could not detect the arise of 1370 cm⁻¹ and 1636 cm⁻¹ upon omeprazole induction. We think the reason is that the basal expression of 1A2 is low. Although several folds of increase can be detected after OME treatment, this amount of CYP induction is still low to be detected by Raman microscopy. But since we can detect the same peaks in DMSO treated HepaRG cells, which universally induce all types of CYPs. We think the multiple CYPs will be more precise. Therefore, we made changes as the following, which can be found in Lines 1-11 on Page 5.

“These results suggest that the increase of Raman shifts at 1370 cm⁻¹ and 1636 cm⁻¹ can be associated with the induction of CYP3A4 by RIF. However, although RIF is a strong CYP3A4 inducer, other CYPs, like 2B6, was also elevated in HepaRG cells when treated with RIF (Supplementary Fig. 1). Since the heme b core is the same in all CYP subtypes, other CYP types should also contribute to the arise of Raman peaks at 1370 cm⁻¹ and 1636 cm⁻¹. We also treated HepaRG cells with DMSO and omeprazole, a universal CYP inducer and a CYP1A2 inducer, respectively. The significant increase in 1370 cm⁻¹ and 1636 cm⁻¹ was revealed in DMSO-treated cells but not omeprazole (Supplementary Fig. 2). Since the same Raman shifts were detected in DMSO-treated cells, the Raman shifts at 1370 cm⁻¹ and 1636 cm⁻¹ can be assigned to several CYPs including CYP3A4. Although the treatment with OME elevated the expression of 1A2 for several folds, this amount is still undetectable by Raman microscopy due to the possible low basal expression of CYP1A2 in HepaRG cells (Supplementary Fig.2).^{29,35”}

8. *The inducer utilized in this paper may influence multiple CYPs. RIF could not only induce the CYP3A4 expression, but also CYP2C9 and so on. Other CYP3A4 inducers in this paper may influence multiple CYPs as well. For example, Phenytoin is a known inducer of drugs metabolized by CYP2C, CYP2D, and CYP3A.*

We thank the reviewer for this comment. Indeed, measuring CYP3A4 activity is not sufficient since other CYP subtypes may also be elevated. Therefore, we redid the enzyme activity of other CYP subtypes together with 3A4, including 1A2, 2B6, and 2C9. This result was added as Supplementary Fig. 7 as mentioned above. As the reviewer mentioned, the treatment of RIF, PHY, CBZ, DEX does not solely increase the activity of CYP3A4, but also other CYP types. Since the elevation of CYP activity is more significant in CYP3A4 and the basal amount of CYP3A4 is more than other types. This result indicates that the increase of CYP Raman shifts is mainly due to the increase in CYP3A4. However, we could not eliminate the contributions of other increased CYPs in our observations. Therefore, we changed our claim of observation CYP3A4 to all CYPs.

We made changes in the description of results of various inducers as follows and also can be found in Line 18-22 on Page 9.

“We also investigated the universal usage of Raman measurements on other strong CYP3A4 inducers (Fig. 3f, j), including phenytoin (PHY), dexamethasone (DEX), and carbamazepine (CBZ), all of which showed the similar tendency as that seen in the luminescence assay of CYP3A4 (Fig. 3f). The induction of other CYP1A2, CYP2B6, and CYP2C9 was shown in Supplementary Fig. 7, indicating that the change in CYP3A4 is dominant.”

When we performed this experiment again, we found that CBZ showed a slightly higher CYP3A4 activity than DEX. We also confirmed this result twice, so we replaced this data in Fig. 3f. Raman data demonstrated a similar value with large sample difference when we compared DEX with CBZ.

9. In the quantitative comparison, the author adopted $I_{1000\text{cm}^{-1}}/I_{1000\text{cm}^{-1}}$ (As shown in the figure). But what is the rationale that the peak intensity at 1000 cm^{-1} to be constant.

Thank you for pointing this out. We used 1000 cm^{-1} as an internal reference to compare among different samples. 1000 cm^{-1} can be assigned to phenylalanine, more specifically, the benzene group in phenylalanine. Since phenylalanine in all proteins, we considered that the intensity of 1000 cm^{-1} represented the whole protein contents in cells. So we use it as an internal reference of cell spectrum. Normalization to phenylalanine is also used by many previous reports. We added one report as a reference (61) in the manuscript and also attached here.

C. Krafft, T. Knetschke, R.H. Funk, R. Salzer. Studies on stress-induced changes at the subcellular level by Raman microspectroscopic mapping. *Anal. Chem.* **2006**, 78, 4424-4429.

10. Fig.3. speculated that CYP inhibition after azamulin treatment led to the Raman shifts observed in HepRG cells. The hypothesis is, active CYPs maintain a ferric low-spin state by using water as a ligand. After azamulin enters the enzymatic pocket, water is replaced by azamulin, which causes low-spin to high-spin transition. It would be great if the author could use sub-cellular system to further approve their hypothesis, for example, CYP3A4 overexpressed microsomes/proteins, which should be commercially available.

We appreciate this suggestion and performed the Raman measurement on corning Supersome overexpressed human CYP3A4 (Cat. No. 456207). The peaks can be assigned to CYPs, including 1370 cm^{-1} and 1636 cm^{-1} , were detected in control supersomes. We also compare the microsomes treated with or without $10\text{ }\mu\text{M}$ Azamulin. A shift from 1636 cm^{-1} to 1621 cm^{-1} was detected. 1636 cm^{-1} is a widely-used marker band for ferric low-spin state, while 1621 cm^{-1} can be assigned to high-spin heme. Other low-spin marker bands at 1590 cm^{-1} and 1500 cm^{-1} are not detectable due to 532 nm excitation or overwhelmed in cell spectrum. Therefore, in HepaRG cells, we only use 1636 cm^{-1} to indicate spin state change. On the other hand, a slight shift from 1370 cm^{-1} to 1367 cm^{-1} was detected as well as the intensity. Bands not sensitive to spin-state transition, such as 750 cm^{-1} , showed no change after azamulin treatment. By using Raman imaging, the changes at different wavenumbers were observed simultaneously. Therefore, microsome study can be another application using Raman microscopy. We added this result as Supplementary Fig. 10 and added the description as follows, which can be seen from Line 27 on Page 12 to Line 4 on Page 13.

“We also performed the inhibition experiment using CYP3A4 overexpressed microsomes. Microsomes were treated with $10\text{ }\mu\text{M}$ AZA for 10 min , and the Raman spectra were collected (Supplementary Fig. 10). We detect the transition from 1636 cm^{-1} to 1621 cm^{-1} and even shift from 1370 cm^{-1} to 1367 cm^{-1} as well as the decrease in intensity of 1636 cm^{-1} and 1370 cm^{-1} . In living cell condition, Raman shifts around 1621 cm^{-1} was overwhelmed by other cellular peaks and 1370 cm^{-1} was affected by the adjacent peak at 1360 cm^{-1} . Therefore, 1636 cm^{-1} is more useful to evaluate CYP inhibition in HepaRG cells.”

Supplementary Fig. 10 Raman spectra of microsomes and the effect of AZA on spectrum. **a** The reconstructed Raman image of microsomes at 750, 1370, 1636, and 1621 cm^{-1} with and without AZA treatment. Scale bars, 20 μm . **b** The Raman spectra of microsomes treated with and without AZA. LS marker band at 1636 cm^{-1} decreased after AZA treatment while HS marker band at 1621 cm^{-1} increased, demonstrating the LS to HS transition caused by AZA treatment. Meantime, the Raman intensity at 1370 cm^{-1} was decreased, and a shift to 1367 cm^{-1} was observed. Since 750 cm^{-1} is not sensitive to spin state transition, the intensity of 750 cm^{-1} did not vary after AZA treatment.

Minor comments:

11. In the HepaRG images (Fig. S2), in order to identify hepatocytes and biliary cells, the related biomarkers (by immunofluorescence staining) should be used to characterize the two kind of cells, not only use Raman spectra.

Immunofluorescence staining results of CK-19 (biliary marker) and CYP3A4 (hepatocyte marker) were added into Fig. 2b and also shown here. Hepatocyte colony express CYP3A4 while the surrounding biliary cells do not express CYP3A4 but express CK-19 protein.

Fig. 2 Hepatocyte specific response to CYP induction was monitored by Raman microscopy. a The Raman spectra of hepatocytes (red trace) and biliary cells (gray) without CYP induction. **b** Immunofluorescence staining of CYP3A4 (green) and CK-19 (red) with nuclei counterstaining by DAPI. Scale bar, 50 μm . Reconstructed Raman images of control hepatocytes (Ctrl-H, **c**), control biliary cells (Ctrl-B, **d**), RIF-treated hepatocytes (RIF-H, **e**), and RIF-treated biliary cells (RIF-B, **f**) demonstrate the hepatocyte-specific response to RIF induction. Scale bars, 20 μm . **g** The average spectra of Ctrl-H (red), Ctrl-B (gray), RIF-H (light red), and RIF-B (light gray). 1370 cm^{-1} and 1636 cm^{-1} were indicated to show the appearance only in RIF-H.

Additionally, fluorescent image of *cyt b₅* at the same position as the Raman image were also added as Supplementary Fig. 4. It is clearly shown that hepatocyte colony expressed *cyt b₅* much more than biliary cells.

Supplementary Fig. 4 Reconstructed Raman images of Hepatocyte and biliary cells and in situ immunofluorescence staining of *cyt b₅*. A The Raman images at 600, 1000, 675, and 940 cm^{-1} , which can be assigned to *cyt c*, phenylalanine, *cyt b₅*, and glycogen, respectively. Scale bars, 20 μm . **b** Immunofluorescence staining of *cyt b₅* with nuclei counterstaining by DAPI. Scale bars, 20 μm . The staining images indicate the *cyt b₅* is expressed in hepatocytes but not biliary cells.

- 12. Raman spectroscopic characterization of hepatocytes and biliary cells in HepaRG culture showed significant differences (Fig. S2). The CYP inducer/inhibitor impacts on the biliary cells Raman signals are critical to evaluate the non-invasive method. It would be great if the author could compare the spectra in both cells instead of just hepatocyte-like cells. Therefore, Fig. S4 should be moved to the draft instead of in the supplemental part.**

We very appreciate this suggestion and have arranged Fig. S4 as Fig. 2 to state hepatocyte specific response to CYP induction, as seen in our answer to the previous question. We also added the detailed description on Raman spectroscopic characterization of hepatocytes and biliary cells as well as the hepatocyte specific response to CYP rifampicin. The newly added information is shown as follows, and also can be found in Lines 1-25 on Page 7.

“Non-invasive detection of hepatocyte-specific response to CYP induction

Non-invasive characterization of different cell populations in an in vitro model could pave the way for the evaluation of cell type-specific and subsequent intercellular responses after drug administration. Differentiated HepaRG cells were known to present two morphologically and functionally distinct cell populations, including hepatocytes and biliary cells.⁴³ Hepatocytes form colonies and are surrounded by flat biliary cells. Raman spectra of hepatocyte colonies and biliary monolayer were collected in HepaRG cell without induction. The results showed clear differences in the Raman spectra of the two cell populations (Fig. 2a). The Raman spectra of biliary cells were dominated by resonance peaks from reduced cyt c (600 cm^{-1} , 640 cm^{-1} , 750 cm^{-1} , 1130 cm^{-1} , 1313 cm^{-1} , and 1585 cm^{-1}) and protein (1000 cm^{-1} , phenylalanine) and lipid (1450 cm^{-1} , CH_2 deformation) bands. In contrast, hepatocytes (without induction) preserved all the above-mentioned Raman shifts with higher intensities, indicating that these biomolecules are more abundant in hepatocytes. Furthermore, Raman shifts corresponding to reduced b-type cytochrome (675 cm^{-1} and 1304 cm^{-1}) and glycogen (940 cm^{-1} and 1084 cm^{-1}) appeared in hepatocytes. A comparison of representative 675 cm^{-1} and 940 cm^{-1} peaks is shown in the inset of Fig. 2a. The reconstructed Raman images comparing hepatocytes and biliary cells were shown in Supplementary Fig. 4. The immunostaining images of cyt b₅ at the same position of Raman observation indicated that more cyt b₅ is expressed in hepatocytes than biliary cells (Supplementary Fig. 4b).

After RIF treatment, expression of CYP3A4 was confirmed in the hepatocyte region by immunostaining (Fig. 2b, green). The surrounding biliary cells were positive in CK-19 staining, a biliary marker (Fig. 2b, red). The reconstructed images at 600 cm^{-1} , 1370 cm^{-1} , and 1636 cm^{-1} were shown to reveal the localization of cyt c, oxidized CYPs and LS CYPs. The results indicated that hepatocyte region showed a clear cell contrast of 1370 cm^{-1} and 1636 cm^{-1} , whereas only mitochondrial contrasts were observed in biliary cells, indicating that the 1370 cm^{-1} and 1636 cm^{-1} occurrences were more specific to hepatocytes (Fig. 2c). This change was also clearly visible in the comparison of Raman spectra (Fig. 2d). This also indicates that Raman microscopy is capable to monitor CYP induction. In our experiment, without any labeling, distinct Raman signatures of hepatocytes and biliary cells were characterized, enabling the study of cell-type-specific drug responses.”

- 13. In general, CYP induction would be conducted by 72h inducing. Please list the rationale why only 48h was used in this study.**

We followed the protocol provided by Biopredic International that “Maximal fold induction of metabolic activity may be achieved with 72 hours treatment time, but vendor’s data indicate that 48 hours of treatment is sufficient to demonstrate significant induction of CYP1A2, CYP2B6, and CYP3A4 metabolic activity using prototypical inducers.” A condition which can demonstrate significant induction will satisfy our purpose, so we chose 48h.

14. For CYP inhibition, why it was conducted after 48h induction? In general, it should be conducted in the cells without inducing treatment.

Without inducing treatment, HepaRG cells express very low amount of CYP. We should use either DMSO or inducers like RIF to elevate the expression of CYP at first. Since Azamulin is a specific CYP3A4 inducer, we treated cells with RIF for 48 h and then treated with Azamulin. We think in this way we can closely approach to CYP3A4 activity more specifically.

15. In this paper, each high-resolution Raman image contains 400 pixels x 252 pixels for a total of 100, 800 spectra per image. What is the integration time for each spectrum and how long it may take to obtain such a high-resolution image?

Thank you for asking this question. We used a homebuilt line-illumination Raman confocal microscope. The laser light is shaped into a line, and 400 spectra were obtained at this single line simultaneously (Palonpon et al., *Nat. Protoc.* **8**, 677-692, 2013). Compared with the conventional point-scan Raman microscopy the imaging speed was elevated several hundred times, thus enabling live cell imaging. The exposure time of one line is 5s and integration time of signal is about 4.6 s. Totally, to obtain 400 spectra, 9.6 s is required. To obtain high-resolution image (Figure 1, 3 and 4) containing 400 pixels x 252 pixels, around 40 min (9.6 s x 252 lines) is necessary. For spectrum analysis, such a high-resolution image is not required, so we can decrease the resolution by increasing step size. For the quantitative analysis of Raman intensity, we only scanned 84 lines instead of 252 lines at a step size of 1 μm . The images shown in Fig. 3 are the low-resolution examples. In this case, 13 min (9.6 s x 84 lines) is required for one image. In our previous studies, time-lapse observation using this Raman microscope was performed and dynamics of cytochrome c in HeLa cell division (Hamada K et al., *J. Biomed. Opt.* 2008) and cell apoptosis (Okada et al., *PNAS* 2011) was monitored without labels. Our attempt on hepatocyte unveiled more hepatic molecules (CYP, glycogen, and carotenoids) than HeLa cells and further expanded the biological application of Raman microscopy.

16. The hyperspectral Raman images in this paper should provide their brightness scale.

We apologize for missing this information. We have added the brightness scale for all Raman images we provided, please see changes in Fig. 1, 2, 3, 4, 5 and Supplementary Fig. 3, 4, 10, 12.

Reviewer #3 (Remarks to the Author):

The authors of the manuscript "Label-free chemical imaging of cytochrome P450 activity by Raman microscopy" present studies in which they use Raman microscopy to visualize and evaluate the activity of cytochrome P450, more specifically CYP3A4, under the physiological condition in human hepatocytes. CYP3A4 is a crucial cytochrome P450 in human physiology since it is responsible for the metabolism of almost 50% of human drugs. Raman microscopy is a noninvasive, label-free technique that has great potential for studying critical physiological processes at the cellular level. The authors convincingly argue that the CYP distribution within the cells can be successfully monitored using Raman microscopy, and even more significantly, allow probing some of their activity. The authors support their spectroscopic findings with a broad array of other techniques, such as activity measurements, immunofluorescence staining, Western blotting, and statistical analysis. As such, I think the presented overall findings are of great importance and will be of great interest to the broad scientific community. However, while the general conclusions are correct, meaning that the authors present convincing arguments that they can monitor CYP in the cells and utilize the Raman microscopy to detect CYP activity under various conditions, there are several major problems, in my opinion, with data interpretation, that needs to be addressed before publication in *Communications Biology*.

Major problems:

17. The authors interpretation of Raman data in Figure 1.a that led to the statement: ferric CYPs were the functional form in living human hepatocytes; (page 5, lines 11-12) is, in my opinion, incorrect. I agree that the cell induction results in the increase of the 1370 cm^{-1} band (Figure 1.a), which indicates that the protein exists in a ferric state (*Biochemistry*, 1992, 31, 4384-4393). However, there is an additional band

of equal intensity in this spectrum at around 1360 cm⁻¹, which is close to the 1345 cm⁻¹, characteristic of ferrous state (Biochemistry, 1992, 31, 4384-4393), meaning that the protein exists as a mixture of ferric and ferrous states. Actually, the authors do recognize that this band is associated with the reduced form (page 7, line 5). Please also note that this band is dominant in the spectrum of control protein, indicating that the protein before induction is mainly in the ferrous state. I recommend that the authors rewrite this paragraph and focus on making the point that they are able to convincingly visualize CYP using a 1370 cm⁻¹ oxidation band without concluding that the ferric form is the only form of CYPs in living cells.

We thank the reviewer for this comment and apologize for the confusion. We noticed that 1360 cm⁻¹ existed in control HepaRG cells, and it is constant even after rifampicin treatment, which means this content does not respond to CYP induction. Therefore, we think the assignment of 1360 cm⁻¹ is not from CYPs, but other heme b protein, like abundant cyt *b*₅ in hepatocytes. However, we cannot eliminate the possible ferrous CYP in HepaRG cells. We modified the text based on the reviewer's comments and state we can visualize induced CYP using 1370 and 1636 cm⁻¹ band and remove the claim that the ferric form is the only form of CYPs in living cells. Please see changes in the manuscript as follows and also can be seen in Lines 14-17 on Page 4.

“From the previous resonance Raman studies of the CYP protein, substrate binding to CYPs at the ferric and low-spin (LS) resting state and triggers the CYP catalytic cycle. 1370 cm⁻¹ and 1636 cm⁻¹ were generally used as an oxidation and LS state marker of CYP, respectively.^{25,27,31} The appearance of 1370 cm⁻¹ and 1636 cm⁻¹ peak after RIF treatment may suggest that the induced CYP is at an oxidized and LS state.”

Others include:

Lines 6-8 on Page 2

“The redox-state and spin-state sensitive Raman measurement indicated that the induced CYPs in living hepatocytes were in the oxidized form at low-spin state, which is related to the monooxygenase function of CYP”

From Line 28 on Page 5 to Line 2 on Page 6.

“Using Raman microscopy, we directly visualized cellular CYPs without external labels and demonstrated that ferric CYPs with LS were the major form upon the RIF induction in living HepaRG cells.”

18. The paragraph describing mechanism-based inhibition needs major revision:

a The 1370 cm⁻¹ band is an oxidation state marker, not a spin state marker. The addition of azamulin does not change the oxidation state but the spin state. As such, including the 1370 cm⁻¹ band as an indicator of mechanism-based inhibition is incorrect. I do agree, though, that the 1636 cm⁻¹ mode can be indicative of spin state change associated with the binding of azamulin. The transition from 6 coordinated low spin (6cLS) to 5 coordinated high spin (5cHS) state, which is typically reflected by nu10 mode shifting from around 1640 cm⁻¹ (for 6cLS) to 1620 cm⁻¹ (for 5cHS) (Biochemistry, 1992, 31, 4384-4393) which can be observed in the spectrum as lowering of the 1636 cm⁻¹ mode. Please rewrite this part accordingly.

We really appreciate this comment and realize that we should separately discuss 1370 cm⁻¹ as an oxidation marker and 1636 cm⁻¹ as a spin state marker. So, we use the decrease of 1636 cm⁻¹ to indicate the LS to HS transition in the revised manuscript.

Also, we would like to discuss about nu4 mode at 1370 cm⁻¹. Although 1370 cm⁻¹ is not a considered spin state marker, we find that a three-wavenumber downshift was observed in supersome treated with azamulin (Supplementary Fig. 10 and also used to answer Question 10 from the 2nd reviewer.). The intensity of 1370 cm⁻¹ was also decreased. Since it is a generally accepted oxidation marker rather than spin state marker, we revised the manuscript and showed below.

The above discussion was included in the revised manuscript in Lines 19-26 on Page 12.

“Three spin state marker bands are sensitive to the changes in porphyrin core size caused by the transition of spin state. These bands were found around 1500 cm⁻¹ (ν3), 1590 cm⁻¹ (ν2), and 1640 cm⁻¹ (ν10) for the LS state and around 1490 cm⁻¹, 1570 cm⁻¹, and 1620 cm⁻¹ for the HS state.^{25,27,31} In our measurement, 1500 cm⁻¹ and 1590 cm⁻¹ are not detectable due to the contamination of cellular components or the possible lower efficiency of Raman scattering at 532 nm. We observed a decrease of 1636 cm⁻¹ which allowed us to evaluate inhibition in living cells using this peak. Although 1370 cm⁻¹ is not a so-called spin state marker, the decrease in intensity was observed in our measurement (Fig. 4b), which might be associated with the reduction of CYP after binding with AZA.”

b Figure 3.a – The schematic drawing of azamulin as coordinated to the heme is implausible. The authors of reference 44 showed that the azamulin binds within the heme pocket and replaces the water cluster, which triggers a switch from 6cLS state to 5cHS state (see Figure 3, ref. 44). Coordination of azamulin to the heme iron (as the Figure 3.a suggests) would most likely result in different UV-Vis response and produces a 6cLS state with a Soret maximum around 420 nm, which is not observed (ref 44. Figure 3). Furthermore, the sentence on page 9, lines 4-5 is not accurate. Binding to the heme core (meaning porphyrin macrocycle or its peripheral groups) is only one of many ways of mechanism-based inhibition; others are binding to the heme iron or active site amino acid residues. Please rewrite the sentence.

We apologize for this confusion and realize that Fig. 3a is not correct because it did not indicate the 6cLS to 5cHS transition. Based on ref. 44, the binding of Azamulin to the active site was associated with multiple van der Waals contacts and two hydrogen bond between Azamulin and apoprotein. Therefore, we modified Fig. 3a (now Fig. 4a) to show 6cLS to 5cHS transition. Also, the description of mechanism-based inhibition is also not precise, so we modified this part according to the reviewer's comment. Please see changes as following, which can be found in Lines 4-5 on Page 12, and the answer to the previous question.

“The catalysis intermediate binds with the heme core or apoprotein and blocks the entrance of other substrates to irreversibly inactivate the enzyme”

Minor issues:

19. Page 3, line 14; needs reference

Thank you for this comment. We labeled reference 9-11 for this sentence.

*“9. K. A. Youdim, K. C. Saunders, A review of LC–MS techniques and high-throughput approaches used to investigate drug metabolism by cytochrome P450s. *J. Chromatogr. B.* **2010**, 878, 1326-1336.
10. J. Wu, X. Guan, Z. Dai, R. He, X. Ding, L. Yang, G. Ge, Molecular probes for human cytochrome P450 enzymes: Recent progress and future perspectives. *Coord. Chem. Rev.* **2021**, 427, 213600.
11. J. J. Cali, D. Ma, M. Sobol, D. J. Simpson, S. Frackman, T. D. Good, W. J. Daily, D. Liu, Luminogenic cytochrome P450 assays. *Expert Opin. Drug Metab. Toxicol.* **2006**, 2, 629-645.”*

20. Page 3, line 117-18 sentence “The functions of catalytic” is unclear, I am not sure what the authors mean

We meant that the direct impact on CYP itself is hard to clarify if the final metabolites are used for evaluation, because it is difficult to eliminate the possible alterations on associating factors, like redox partner or drug transporters. We realized that the “the function of catalytic” is confusing, so we modified the sentence. Please see changes as follows, and also can be found in Line 16-17 on Page 3.

“Whether the impact is directly on CYP itself or other associating factors is difficult to clarify.”

21. Page 4, lines 15-16 which were assigned to oxidized heme, please be more specific, modes around 1370 cm⁻¹ are associated with nu4, which is the oxidation state of heme proteins; however modes at around

1636 cm^{-1} are associated with vinyl stretching modes and/or nu10 mode which is a heme iron spin state marker (not oxidation state marker).

Thank you for pointing out this issue. We have corrected the description based on this comments. Please see changes below and also in Lines 15-17 on Page 4.

“1370 cm^{-1} and 1636 cm^{-1} were generally used as an oxidation and LS state marker of CYP, respectively.^{25,27,31} The appearance of 1370 cm^{-1} and 1636 cm^{-1} peak after RIF treatment may suggest that the induced CYP is at an oxidized and LS state.”

- 22. It is possible (and quite likely) that the inducers or downregulator exhibit their own Raman spectrum; how did the authors account for the possible contribution of the inducers/downregulator scattering to their data analysis (page 7 and 8)**

Thank you for raising this issue. We have measured the Raman spectra of stock solutions of RIF (20 mM), PHY (500 mM), CBZ (500 mM), DEX (500 mM), IL-6 (10 ug/ml) as well as OME (200 mM) and AZA (10 mM). The solvent of RIF, PHY, CBZ, DEX, OME, and AZA is DMSO, while the solvent for IL-6 is PBS. The Raman spectra of these stock solutions are shown below. No characteristic peaks of these chemicals were detected at the target wavenumbers of 1370 cm^{-1} and 1636 cm^{-1} . Therefore, the possible contribution of the inducers/downregulator/inhibitor scattering can be considered ignorable. We realized that this information is very important, therefore, we inserted Figure R?this figure as Supplementary Fig. 7 in the supporting information and added the descriptions in the manuscript as follows, which can be found also in Lines 6-9 on Page 10.

“To eliminate the possible contamination of Raman scattering from inducers or down-regulators. The Raman spectra of stock solution of each chemical were shown in Supplementary Fig. 8. No overlapping at 1370 cm^{-1} and 1636 cm^{-1} as well as no characteristic peaks of these inducers and down-regulators were found in cell spectrum.”

Supplementary Fig. 8. Raman spectra of inducers, down-regulators and inhibitors. (A) RIF. (B) PHY, DEX, CBZ, OME, AZA, and DMSO (solvent). (C) IL-6 and PBS (solvent).

23. What is the spectral resolution of reported Raman spectra?

The spectral resolution used in this manuscript is 1.5 cm^{-1} . We also added this information in Lines 16-17 on Page 20.

“The spectral resolution was 1.5 cm^{-1} .”

Reference

- S. Kawata, R. Arimoto, O. Nakamura, Three-dimensional optical-transfer-function analysis for a laser-scan fluorescence microscope with an extended detector. *J. Opt. Soc. Am. A.* **1991**, 8, 171-175.
K. Hamada, K. Fujita, N. I. Smith, M. Kobayashi, Y. Inouye, S. Kawata, Raman microscopy for dynamic molecular imaging of living cells. *J. Biomed. Opt.* **2008**, 13, 044027.
M. Okada, N. I. Smith, A. F. Palonpon, H. Endo, S. Kawata, M. Sodeoka, K. Fujita, Label-free Raman observation of cytochrome *c* dynamics during apoptosis. *Proc. Natl. Acad. Sci. U. S. A.* **2012**, 109, 28-32.

Finally, we thank the Editor and the reviewers for giving us the opportunity to strengthen our manuscript with the valuable comments and queries. We have worked hard to incorporate the feedback and hope that our revised submission is now acceptable for publication in *Communications Biology*

Sincerely yours,

Katsumasa Fujita, Ph. D.
Professor, Department of Applied Physics, Osaka University
2-1 Yamadaoka, Suita, Osaka, 565-0871, Japan
Tel.: +81-6-6879-7847
Email: fujita@ap.eng.osaka-u.ac.jp

REVIEWERS' COMMENTS:

Reviewer #1 (Remarks to the Author):

532nm Raman is widely used for in-vitro or ex-vivo applications (ex. tissue slice). It will be helpful to add couple of references for 532nm in-vivo applications.

Other than that, raised concerns are well addressed.

Appreciate the additional experiment to characterize the axial resolution of the slit scanning system.

Reviewer #2 (Remarks to the Author):

The authors have answered most of the questions I addressed before. The results are solid and I don't have further comments.

Reviewer #3 (Remarks to the Author):

The authors addressed all my concerns satisfactorily, therefore I recommend this manuscript for publication.

REVIEWERS' COMMENTS:

Reviewer #1 (Remarks to the Author):

532nm Raman is widely used for in-vitro or ex-vivo applications (ex. tissue slice). It will be helpful to add couple of references for 532nm in-vivo applications. Other than that, raised concerns are well addressed. Appreciate the additional experiment to characterize the axial resolution of the slit scanning system.

Thank you very much for this suggestion. We added a sentence related to in-vivo applications in Lines 26-27 on Page 3 and shown below.

“Also, Raman microscopy have been widely used in tissues *in vivo* or *ex vivo* for disease diagnosis.²⁵⁻²⁷”

References 25-27 were added as the reviewer recommended.

25. M. Ogawa, Y. Harada, Y. Yamaoka, K. Fujita, H. Yaku, T. Takamatsu, Label-free biochemical imaging of heart tissue with high-speed spontaneous Raman microscopy. *Biochem. Biophys. Res. Commun.* **2009**, 382, 370-374.
26. W. Muller, M. Kielhorn, M. Schmitt, J. Popp, R. Heintzmann, Light sheet Raman micro-spectroscopy. *Optica* **2016**, 3, 452-457.
27. B. Lochocki, B. D. C. Boon, S. R. Verheul, L. Zada, J. J. M. Hoozemans, F. Ariese, J. F. de Boer. Multimodal, label-free fluorescence and Raman imaging of amyloid deposits in snap-frozen Alzheimer's disease human brain tissue. *Commun. Biol.* **2021**, 4, 474.

Thank you very much for reviewing our manuscript.

Reviewer #2 (Remarks to the Author):

The authors have answered most of the questions I addressed before.
The results are solid and I don't have further comments.

We greatly appreciate the comments from the reviewer which help to explore more applications in liver research using Raman microscopy. We hope to have further discussions.

Reviewer #3 (Remarks to the Author):

The authors addressed all my concerns satisfactorily, therefore I recommend this manuscript for publication.

We thank the reviewer for the comments in terms of protein Raman spectroscopy. The remarks have strengthened our manuscript. We hope to have further discussions.

Finally, we thank the Editor and the reviewers for giving us the opportunity to strengthen our manuscript with the valuable comments and queries. We hope that our revised submission is now acceptable for publication in *Communications Biology*

Sincerely yours,

Katsumasa Fujita, Ph. D.

Professor, Department of Applied Physics, Osaka University

2-1 Yamadaoka, Suita, Osaka, 565-0871, Japan

Tel.: +81-6-6879-7847

Email: fujita@ap.eng.osaka-u.ac.jp